# Teaching Metric Distance
# to Discrete Autoregressive Language Models

**Jiwan Chung[1], Saejin Kim[3], Yongrae Jo[2], Jaewoo Park[1], Dongjun Min[1], Youngjae Yu[3]**
[1]Yonsei University    [2]LG AI Research    [3]Seoul National University
jiwan.chung.research@gmail.com

## Abstract

Large language models (LLMs) operate as autoregressive predictors over discrete token vocabularies, a formulation that has enabled their adaptation far beyond natural language to vision, robotics, and multimodal reasoning. However, training against one-hot targets disregards metric relationships between tokens and limits effectiveness on tasks where distance is meaningful, such as numerical values, spatial coordinates, or quantized embeddings. We introduce DIST$^2$Loss, a distance-aware objective for discrete autoregressive models that replaces one-hot targets with reward-weighted distributions derived from predefined token distances. DIST$^2$Loss can be interpreted as the closed-form solution to entropy-regularized policy optimization with known per-token rewards, retaining the core mechanism of reinforcement learning while avoiding sampling, rollouts, and instability. Our experiments show that DIST$^2$Loss improves data efficiency and downstream performance across diverse domains. It yields tighter bounding boxes in visual grounding, accelerates robotic manipulation by improving action learning, enhances reward modeling for LLM alignment, and strengthens vector-quantized image generation. These results demonstrate that distance-aware supervision offers a simple and general alternative to one-hot supervision for discrete autoregressive models.

## 1 Introduction

Large language models (LLMs) (Radford et al., 2018) have recently emerged as backbones for general-purpose foundational models across wide domains (Bommasani et al., 2021). These models rely on two probabilistic principles. First, they represent a sample text as a sequence of tokens and train the model *autoregressively*, predicting each token conditioned on the previous ones. Second, each token is treated as a *discrete* categorical variable, optimized to match a one-hot target distribution.

While originally developed for natural language, LLMs are now widely adapted to tasks far beyond text. In vision, they have been coupled with discrete visual tokens for image generation and editing (Esser et al., 2021; Dhariwal et al., 2020); in robotics, they are finetuned to handle control and planning tasks by treating actions or trajectories as token sequences (Xiao et al., 2024; Li et al., 2024); and in multimodal reasoning, they are adapted to align visual, textual, and symbolic representations (Yu et al., 2024). These cases demonstrate the portability of the discrete autoregressive formulation beyond language.

A key limitation of such adaptation is the inability to fully exploit numerically or metrically structured elements. These include explicit numbers, as well as entities situated in broader *metric spaces*, such as integers in arithmetic reasoning (Yuan et al., 2023), spatial coordinates and rotation angles in object detection and manipulation (Xiao et al., 2024; Li et al., 2024), and high-dimensional quantized embeddings in image or video generation (Esser et al., 2021; Yu et al., 2024; Dhariwal et al., 2020). In conventional finetuning, the intrinsic distance structure among these elements is ignored, since tokens are reduced to one-hot categorical targets.

In this work, we introduce DIScreTized DISTance Loss (DIST$^2$Loss), a framework that integrates predefined distance relationships between tokens into the adaptation of autoregressive discrete models. DIST$^2$Loss requires no additional data and incurs minimal computational overhead, enabling plug-and-play use across diverse setups. By encoding metric structure directly into the target distribution, DIST$^2$Loss accelerates performance improvement on tasks where distances are semantically

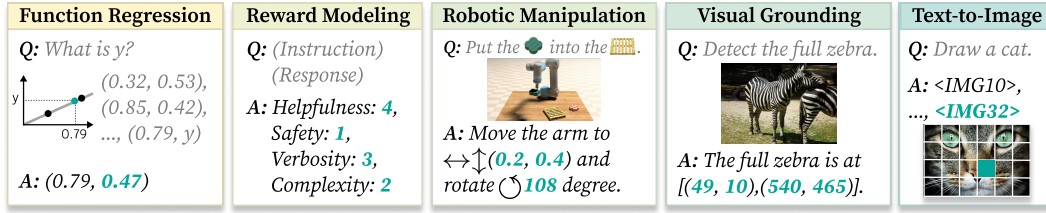

Figure 1: Tasks beyond language frequently involve outputs with inherent metric structure, such as quantities or coordinates, highlighting domains where distance modeling could be beneficial.

meaningful, including object detection (section 3.2), object manipulation (section 3.3), reward modeling (section 3.4), and image generation (section 3.5).

Conceptually, DIST²Loss can be viewed as the closed-form solution to entropy-regularized policy optimization, providing a stable and efficient alternative to reinforcement learning. It constructs a reward-weighted target distribution over the vocabulary and trains the model to match it through KL divergence. This preserves the essential mechanism of reward alignment while avoiding the sampling, rollouts, and instability characteristic of traditional RL methods. Crucially, such rewards are only well defined when tokens admit a meaningful metric, such as numerical values, coordinates, or quantized embeddings, so that distances can be translated into scalar quality signals. In domains without intrinsic geometry, the method reduces to one-hot supervision.

Our experiments demonstrate that DIST²Loss generalizes effectively across domains and improves downstream performance even in data-scarce settings. It yields tighter bounding box predictions in visual grounding (section 3.2), accelerates the learning of robotic actions to increase success rates in manipulation tasks (section 3.3), improves reward modeling for LLM alignment (section 3.4), and enhances the learning of vector-quantized image representations in autoregressive models (section 3.5). These results illustrate that distance-aware supervision can consistently strengthen discrete autoregressive models beyond one-hot next-token prediction.

## 2 METHOD

We aim to design an objective that (1) leverages the given metric to construct optimization targets, and (2) remains compatible with the categorical distributions used in LLM-based foundational models. We hypothesize that incorporating this metric prior improves data efficiency when distances are meaningful. This section is organized as follows: first, we review the conventional cross-entropy formulation; second, we introduce DIST²Loss; third, we interpret DIST²Loss from a reinforcement learning perspective; and finally, we extend the framework to high-dimensional metrics.

### 2.1 PRELIMINARIES

**Notations.** Let $\mathcal{V}$ denote the vocabulary of the foundational model, and consider a subset $\mathcal{V}_d \subseteq \mathcal{V}$ with cardinality $|\mathcal{V}_d| = M$. Define a metric space $(\mathcal{X}, d)$, where each element $x \in \mathcal{X}$ represents a sequence $x = (x_1, \ldots, x_L)$ with $x_i \in \mathcal{V}_d$. The metric $d : \mathcal{X} \times \mathcal{X} \to \mathbb{R}$ assigns a distance $d(x, y)$ between any pair of sequences $(x, y) \in \mathcal{X} \times \mathcal{X}$. This distance $d(x, y)$ is determined by the underlying data structure, such as the Euclidean distance for integers or an embedding distance for multi-dimensional vectors.

Consider the discrete input sequence $s = (s_1, \ldots, s_n)$, representing a sequence of tokens in an autoregressive discrete foundational model. A single forward pass through the model generates logits over the entire vocabulary $\mathcal{V}$ for each token in the sequence:

$$\mathbf{l}_t = f_\theta(s_{<t}), \quad \forall t \in 1, \ldots, n \tag{1}$$

where $\mathbf{l}_t$ represents the logit vector at time step $t$ and $f_\theta$ denotes the model parameterized by $\theta$. These logits $\mathbf{l}_t$ are then transformed into probability distributions over the vocabulary subset $\mathcal{V}_d$ by applying the softmax function:

$$p_\theta(v|s_{<t}) = \text{softmax}(\mathbf{l}_t), \quad v \in \mathcal{V} \tag{2}$$

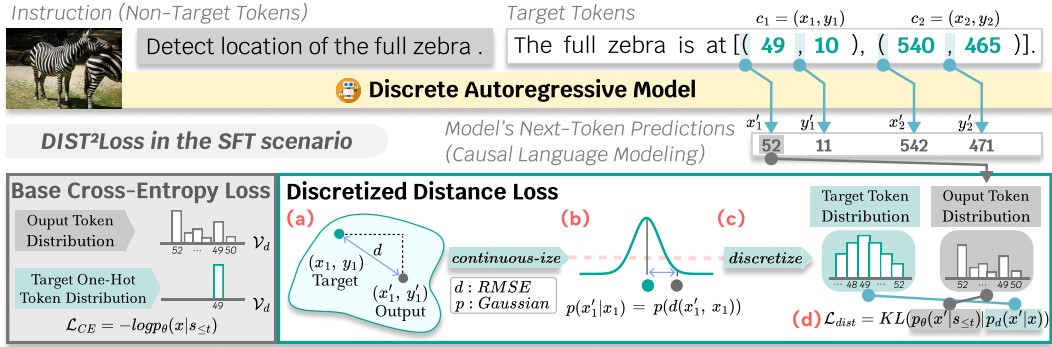

Figure 2: DIST$^2$Loss trains categorical foundational models with a distance-aware target distribution instead of a one-hot target. The procedure is: (a) define a token distance metric $d(x, x')$, (b) convert the metric into a continuous distribution $p(x, x')$, (c) discretize the distribution to obtain $p_d(x, x')$, and (d) compute the KL divergence loss between the target $p_d$ and the model likelihood $p_\theta$ per token.

**Cross-Entropy Loss.** In training a discrete autoregressive model, the standard approach involves teacher-forcing, where the target and model predictions are compared independently at each token. Cross-entropy loss Shannon (1948), $\mathcal{L}_{\text{CE}}$, is commonly used to compare two categorical distributions:

$$\mathcal{L}_{\text{CE}} = -\sum_{t=1}^{n} \sum_{v \in \mathcal{V}} p_{\text{target}}(v|s_t) \log p_\theta(v|s_{<t}) \tag{3}$$

where $p_{\text{target}}(v|s_t)$ denotes the target distribution at time step $t$. In most cases, $p_{\text{target}}(v|s_t)$ is a one-hot distribution that corresponds to the ground truth token $s_t$.

## 2.2 DISCRETIZED DISTANCE LOSS

We define a *structured element* $e$ as the unit on which the metric is defined, such as a bounding box $(x_1, y_1, x_2, y_2)$ or a set of robotic joint angles. For autoregressive modeling, each element is represented as a contiguous subsequence of tokens $x = [x_i : x_j]$ within the input sequence $s = [\ldots, s_{i-1}, x_i : x_j, s_{j+1}, \ldots]$, where each component of the structured object becomes a separate token. In practice, we apply DIST$^2$Loss at each token position $t$ independently, comparing the candidate value $v$ with the corresponding ground-truth component $x_t$. This per-position decomposition is an efficient factorization of the object-level metric: evaluating all multi-token alternatives jointly would scale exponentially with the subsequence length $j - i + 1$. When multiple structured elements $e_1, \ldots, e_K$ appear in a single sequence, their distance-based losses are computed independently and summed.

To incorporate the metric distance into the model's objective, we define a target distribution $p_d(v|x, t)$ that reflects the similarity of the tokens according to a chosen distance metric $d$ in the token space $\mathcal{V}_d$. This target aligns probability mass with the similarity structure, encouraging model outputs that respect the defined metric distance. Crucially, the distance metric $d$ is derived entirely from the task's inherent structure (e.g., Euclidean distance for coordinates, angular distance for rotations, or embedding distance for VQ codes) and requires no additional annotation or supervision beyond the standard ground-truth labels.

We propose formulating the target distribution $p_d$ using a discretized exponential family distribution:

$$p_d(v|x, t) = \frac{\exp\left(-\frac{d(v, x, t)}{\tau}\right)}{\sum_{v' \in \mathcal{V}_d} \exp\left(-\frac{d(v', x, t)}{\tau}\right)} \tag{4}$$

where the temperature hyperparameter $\tau$ controls the smoothness of the target distribution, with lower values of $\tau$ assigning higher probability to tokens closer to the target in the metric space. We set $\tau = 1$ for digit tokens (corresponding to a unit-variance Gaussian) and derive $\tau$ via entropy matching for larger vocabularies such as VQ codebooks; details are provided in section C. Note that in a single

token case, where each element in the metric space consists of a single token from a subset of the vocabulary, we have $d(v, x, t) = d(v, x_t)$ and thus in turn $p_{\text{target}}(v|x, t) = p_{\text{target}}(v|x_t)$.

In the specific case where the root mean squared error (RMSE) is used as the distance metric, this formulation is equivalent to a discretized Gaussian distribution, often referred to as a discrete Gaussian in prior work Canonne et al. (2020). Our framework generalizes this approach, offering a flexible loss function applicable across a range of distance metrics and training setups for foundational models.

The discretized distance loss is defined by comparing $p_{\text{target}}(v|x, t)$ with the model's predicted distribution $p_\theta(v|s_{<t})$ via KL divergence:

$$\mathcal{L}_{\text{dist}} = \sum_{t=1}^{n} \sum_{v \in \mathcal{V}_d} p_{\text{d}}(v|x, t) \log \frac{p_{\text{d}}(v|x, t)}{p_\theta(v|s_{<t})} \tag{5}$$

The final objective combines the cross-entropy loss $\mathcal{L}_{\text{CE}}$ with this distance-based regularization:

$$\mathcal{L} = \mathcal{L}_{\text{CE}} + \alpha \mathcal{L}_{\text{dist}} \tag{6}$$

where $\alpha$ adjusts the weighting between accuracy and metric coherence. For simplicity, we fix $\alpha = 0.1$ throughout the experiments without hyperparameter tuning.

**Example.**  Consider a single-token case, denoted as $x_{\text{single}} \in \mathcal{X}_{\text{single}}$ with $x_{\text{single}} = (x_i)$. To simplify, we restrict the metric space to scalar Euclidean metrics. Suppose the target token $x_{\text{single}}$ is 5, with the Euclidean distance metric defined as $d(v, x) = (v - x_i)^2$. We construct a target distribution $p_{\text{d}}(v|x)$ that assigns higher probabilities to tokens closer to 5 according to this distance. For example, token 4 receives a higher probability than token 2, reflecting its proximity to the target within the metric space. This setup is used directly in our experiments in section 3.4.

## 2.3 CONNECTION TO ENTROPY-REGULARIZED POLICY OPTIMIZATION

The construction of DIST$^2$Loss can be directly linked to entropy-regularized policy optimization. In reinforcement learning, the goal is to optimize a policy $\pi$ over actions $a \in \mathcal{A}$ by maximizing the expected reward. Entropy regularization augments this objective with an entropy term that penalizes peaked distributions and encourages exploration:

$$\max_\pi \ \mathbb{E}_{a \sim \pi}[R(a)] + \tau \mathcal{H}(\pi), \quad \mathcal{H}(\pi) = -\sum_{a \in \mathcal{A}} \pi(a) \log \pi(a).$$

Here, $R(a)$ is the reward associated with action $a$, and $\tau$ is a temperature parameter controlling the strength of regularization. The entropy term prevents the policy from collapsing too early to a deterministic choice and ensures that probabilities remain distributed in proportion to their relative rewards. This objective admits a closed-form optimal policy Haarnoja et al. (2017):

$$\pi^*(a) \propto \exp\left(\frac{R(a)}{\tau}\right).$$

DIST$^2$Loss instantiates this result in the autoregressive modeling setting, where candidate tokens are actions and the reward $R(a)$ is given by a distance-based evaluation. Rather than estimating this distribution through sampling or iterative updates, DIST$^2$Loss uses the analytical solution as the target and trains the model to minimize its KL divergence. Thus, DIST$^2$Loss corresponds to the variance-free, closed-form solution of entropy-regularized reinforcement learning with known per-token rewards.

This interpretation clarifies both the efficiency and the scope of DIST$^2$Loss: it eliminates the instability associated with policy-gradient estimators, but applies cleanly only when rewards are defined independently for each token, such as integers, coordinates, or quantized embeddings.

## 2.4 HIGH DIMENSIONAL DISTANCE

Our DIST$^2$Loss is flexible and can be applied to any distance metric defined over the vocabulary $\mathcal{V}_d$, including the distance between high-dimensional continuous vectors. Here, we outline a practical case where the distance is defined over high-dimensional vector embeddings, which are commonly used in

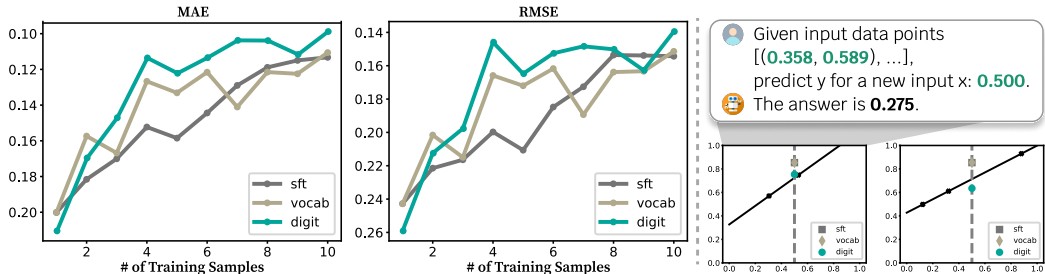

Figure 3: Left: Experimental results showing MAE and RMSE across varying numbers of training samples. The y-axis is inverted for visualization. Right: Overview of the task setup in the meta linear regression experiment, where the model learns to perform linear regression based on the data points.

representation learning Radford et al. (2021); Caron et al. (2021) and information retrieval Karpukhin et al. (2020) literature.

Consider a vector representation $\mathbf{v}(x)$ for each token $x \in \mathcal{V}_d$, where $\mathbf{v}(x) \in \mathbb{R}^D$ is a high-dimensional embedding. Suppose that we have two singleton sequences $x = (x_1)$ and $y = (y_1)$, each represented by their embedding $\mathbf{v}(x_1)$ and $\mathbf{v}(y_1)$. To compute the distance between these sequences, we use a distance metric $d$ over their embeddings, such as cosine similarity or Euclidean distance. For instance, when using cosine similarity, the distance between $\mathbf{v}(x)$ and $\mathbf{v}(y)$ is given by:

$$d\left(\mathbf{v}(x), \mathbf{v}(y)\right) = 1 - \frac{\mathbf{v}(x) \cdot \mathbf{v}(y)}{\|\mathbf{v}(x)\|\|\mathbf{v}(y)\|} \tag{7}$$

which captures the angular separation between token embeddings. The choice of distance metric often depends on the training objective of the embedding function $\mathbf{v}$. For instance, with vector-quantized representations, the distance metric is typically chosen to match the quantization function used during the training of the embedder, as discussed in experiments in section 3.5.

## 3 EXPERIMENTS

We propose a general approach for leveraging metric space information to train discrete foundational models. Our method can be applied whenever a model needs to generate numeric or discretized representations with regression targets. To validate its generality, we apply our approach across a range of tasks: (1) synthetic function regression as a toy task, (2) generative reward modeling for human feedback in LLMs, (3) object detection within multimodal LLMs, (4) object manipulation in embodied AI, and (5) image generation on vector-quantized representations, showcasing its capacity for high-dimensional distance modeling.

**Baselines.** Across our experiments, we evaluate two ablated baselines alongside the full distance-aware loss (*dist*): the *sft* baseline, which applies only the standard cross-entropy loss $\mathcal{L}_{\text{CE}}$ without any distance-specific objective, and the *vocab* baseline, which replaces the distance loss $\mathcal{L}_{\text{dist}}$ with a cross-entropy loss constrained to a subset of the vocabulary $\mathcal{V}_d$. The *vocab* objective is intended to assess the impact of the distance-aware target distribution on model performance, and is defined as:

$$\mathcal{L}_{\text{vocab}} = \mathcal{L}_{\text{CE}}(\mathcal{V}) + \alpha \, \mathcal{L}_{\text{CE}}(\mathcal{V}_d) \tag{8}$$

where $\mathcal{L}_{\text{CE}}(\mathcal{V})$ denotes the cross-entropy loss over the entire vocabulary $\mathcal{V}$ and $\mathcal{L}_{\text{CE}}(\mathcal{V}_d)$ is the cross-entropy loss over the numeric-constrained subset $\mathcal{V}_d$.

### 3.1 TOY: LEARNING TO REGRESS

This experiment represents a *learning-to-learn* task, where the model is trained to acquire the inductive bias of linear regression itself, rather than memorize specific input-output mappings. Each training sample consists of three distinct $(x, y)$ pairs defining a unique linear function, along with a target input $x$ for which the model must predict the corresponding output $y$, as illustrated in fig. 3.

| Models | #PT | #FT | RefCOCO | | | RefCOCO+ | | | RefCOCOg | |
|---|---|---|---|---|---|---|---|---|---|---|
| | | | val | test-A | test-B | val | test-A | test-B | val | test |
| UNINEXT (Yan et al., 2023) | 600K | 127K | 92.6 | 94.3 | 91.5 | 85.2 | 89.6 | 79.8 | 88.7 | 89.4 |
| Ferret (You et al., 2024) | 1.1M | 127K | 89.5 | 92.4 | 84.4 | 82.8 | 88.1 | 75.2 | 85.8 | 86.3 |
| Ferretv2 (Zhang et al., 2024a) | 1.1M | 127K | 92.8 | 94.7 | 88.7 | 87.4 | 92.8 | 79.3 | 89.4 | 89.3 |
| Florence-2-B (Xiao et al., 2024) | 126M | 127K | 92.6 | 94.8 | 91.5 | 86.8 | 91.7 | 82.2 | 89.8 | 82.2 |
| Florence-2-L (Xiao et al., 2024) | 126M | 127K | 93.4 | 95.3 | 92.0 | 88.3 | 92.9 | 83.6 | 91.2 | 91.7 |
| Phi3V (Abdin et al., 2024)-*sft* | 0 | 127K | 94.3 | 93.5 | 86.0 | 85.9 | 91.6 | 78.7 | 92.2 | 87.4 |
| Phi3V-*vocab* | 0 | 127K | 94.5 (↑0.2) | 93.2 (↓0.3) | 86.0 (−) | 85.9 (−) | 90.6 (↓1.0) | 78.2 (↓0.5) | 92.4 (↑0.2) | 87.6 (↑0.2) |
| Phi3V-*dist* | 0 | 127K | 94.8 (↑**0.5**) | 94.5 (↑**1.0**) | 87.3 (↑**1.3**) | 87.1 (↑**1.2**) | 92.2 (↑**0.6**) | 81.4 (↑**2.7**) | 92.8 (↑**0.6**) | 88.0 (↑**0.6**) |

Table 1: RefCOCO (Kazemzadeh et al., 2014; Mao et al., 2016; Yu et al., 2016) visual grounding results (accuracy, %). We fine-tune (FT) Phi3V (Abdin et al., 2024), a model not trained on grounding tasks, while baselines pretrain (PT) on large-scale detection datasets.

Notably, the model is not explicitly informed that the underlying relationship is linear; it must infer this structure from the input data alone.

To evaluate the data efficiency of DIST$^2$Loss, we deliberately restrict the number of training samples to between one and ten, where each sample corresponds to a different regression function with varying slopes and intercepts. This low-data regime is a principled design choice: the goal is not to optimize performance under large-scale supervision, but to assess whether our loss formulation facilitates generalization from minimal structurally meaningful supervision. This aligns with prior work in meta-learning and inductive bias evaluation (Trask et al., 2018; Yu et al., 2020), where models are expected to extract abstract rules from very limited examples.

**Setup.** Each problem is defined by sampling a slope from $[0.1, 1.0]$ and an intercept from $[0.0, 0.5]$. For each random seed, we generate ten training and 1,000 test problems, using subsets of the training set (1–10 samples) to evaluate data efficiency. We fine-tune `meta-llama/Llama-3.2-1B-Instruct` (AI@Meta, 2024) for 5,000 steps using AdamW (batch size 1, learning rate $1 \times 10^{-5}$) with different loss functions, evaluating on unseen regression problems to assess structural generalization. Predictions are made at $x = 0.5$, with performance reported as Mean Absolute Error (MAE) and Root Mean Square Error (RMSE), averaged across five random seeds. All values are reported to three decimal places. Additional details are in section D.2.

**Results.** The bottom panel of fig. 3 demonstrates that DIST$^2$Loss consistently outperforms the baselines (*sft* and *vocab*) in terms of MAE, except when only a single training sample is provided. This exception reflects the challenge of generalizing the linear regression property from a single example. Additionally, the *vocab* baseline shows high variability in regression accuracy across different training data scales due to its tendency to sharpen the target distribution on numerical outputs, leading to inconsistent performance.

## 3.2 MULTIMODAL: VISUAL GROUNDING

We begin by evaluating DIST$^2$Loss on the multimodal task of visual grounding, which involves generating the coordinates of the bounding box for a specified object based on the corresponding referring expression provided as input.

**Setup.** To evaluate data efficiency, we finetune Phi3V[1] (Abdin et al., 2024), which lacks pretrained grounding ability, on RefCOCO (Kazemzadeh et al., 2014; Mao et al., 2016; Yu et al., 2016) without object detection pretraining. Following (Xiao et al., 2024), we combine RefCOCO, RefCOCO+, and RefCOCOg for finetuning. We focus on visual grounding, rather than object detection, to extend LLM language grounding. DIST$^2$Loss is compared against strong baselines pretrained on large-scale grounding datasets, including UNINEXT (Yan et al., 2023), Ferret (You et al., 2024; Zhang et al., 2024a), and Florence-2 (Xiao et al., 2024), all finetuned on the same data. Accuracy (IoU $\geq 0.5$) is the evaluation metric.

**Results.** Table 1 demonstrates that incorporating DIST$^2$Loss consistently enhances performance over the *sft* baseline. In contrast, the *vocab* baseline results varied, underscoring the importance of a

---

[1] `microsoft/Phi-3.5-vision-instruct` (4.2b)

| #Data | L1 | | | L2 | | |
|---|---|---|---|---|---|---|
| | 1K | 10K | 100K | 1K | 10K | 100K |
| RT-2 (Brohan et al., 2023) | 1.9 | 21.9 | 73.1 | 3.8 | 17.7 | 70.4 |
| LLaRA (Li et al., 2024)-*sft* | 49.6 | 82.3 | 88.5 | 46.2 | 78.1 | 84.6 |
| LLaRA-*vocab* | 50.8 | 81.0 | 87.0 | 44.6 | 77.2 | 83.5 |
| LLaRA-*dist* | **53.9** | **83.4** | **89.5** | **51.5** | **82.8** | **86.1** |

Table 2: Object manipulation experiment results on VIMABench (Jiang et al., 2023), reported in accuracy (%). Results are presented for two test protocols (L1 and L2) and various training data scales. For details on baseline scores, refer to section D.3.

| Models | Type | #Data | RewardBench | | | | | MT-Bench |
|---|---|---|---|---|---|---|---|---|
| | | | Chat | Chat Hard | Safety | Reasoning | Average | |
| UltraRM-13B (Cui et al., 2024) | Seq. Classifier | 64K (Cui et al., 2024) | 96.4 | 55.5 | 59.9 | 62.4 | 68.5 | 91.4 |
| Tulu-v2.5-RM-13B (Ivison et al., 2024) | Seq. Classifier | 64K (Cui et al., 2024) | 39.4 | 42.3 | 55.5 | 47.4 | 46.1 | 56.2 |
| Tulu-v2.5-RM-13B (Ivison et al., 2024) | Seq. Classifier | 2M (Ivison et al., 2024) | 93.6 | 68.2 | 77.3 | 88.5 | 81.9 | 91.4 |
| GPT-3.5 (Brown et al., 2020) | Generative | - | 92.2 | 44.5 | 65.5 | 59.1 | 65.3 | 83.3 |
| Claude-3-haiku (Anthropic, 2024) | Generative | - | 73.7 | 92.7 | 52.0 | 79.5 | 70.6 | 82.9 |
| Prometheus-2-7B (Kim et al., 2024b) | Generative | 300K (Kim et al., 2024a) | 85.5 | 49.1 | 77.1 | 76.5 | 72.0 | 75.8 |
| Llama (AI@Meta, 2024)-*binary* | Seq. Classifier | 21K (Wang et al., 2024b) | 83.8 | 34.7 | 39.9 | 73.5 | 58.0 | 62.8 |
| Llama-*sft* | Generative | 21K (Wang et al., 2024b) | 89.1 | 49.3 | 79.2 | 83.9 | 75.3 | 87.3 |
| Llama-*dist* | Generative | 21K (Wang et al., 2024b) | 95.0 (↑**4.9**) | 69.1 (↑**19.8**) | 86.5 (↑**7.3**) | 90.4 (↑**6.5**) | 85.3 (↑**10.0**) | 88.1 (↑**0.8**) |

Table 3: Results of reward modeling experiments on RewardBench (Lambert et al., 2024) and MT-Bench (Zheng et al., 2023), reported in classification accuracy (%). Improvements of DIST$^2$Loss (*dist*) over *sft* are indicated with ↑.

metric-informed target distribution for improved outcomes. With DIST$^2$Loss, Phi3V attains visual grounding performance on par with state-of-the-art models trained on large-scale pretraining datasets optimized for object detection and grounding tasks. Refer to section F for qualitative samples and a hard-case IoU analysis showing that DIST$^2$Loss predictions remain geometrically closer to the ground truth even when incorrect.

## 3.3 Embodied: Robotic Manipulation

Robotic manipulation is another domain where foundational models frequently encounter numerical data. Here, the model must generate robotic joint actions, typically represented by position coordinates and rotation angles, based on contextual inputs and task instructions.

**Setup.** VIMABench (Jiang et al., 2023) is a benchmark for robotic manipulation, encompassing a diverse array of robot arm manipulation tasks organized into 17 distinct categories. It assesses generalization abilities across four levels (L1–L4), with this study focusing on levels L1 and L2. Baseline models include recent multimodal LLM-based approaches, notably RT-2 (Brohan et al., 2023) and LLaRA (Li et al., 2024). This work follows the experimental framework of LLaRA, which fine-tunes the multimodal LLM, LLaVA-1.5[2] (Liu et al., 2024), using instruction-tuning data. Additionally, the scalability protocol from the same study is implemented, where data splits are defined according to dataset size. Consistent with LLaRA's setup, only the loss function is modified, with LLaRA-*sft* serving as a direct baseline. Furthermore, auxiliary tasks introduced in the study are incorporated to expand the training dataset.

**Results.** Table 2 shows a consistent increase in robotic manipulation accuracy with DIST$^2$Loss. Notably, its advantages are pronounced in data-scarce conditions, where training data is limited to approximately 1K samples, further underscoring the effectiveness of the distance metric as a meaningful prior in robotic manipulation learning. Although the performance difference between *dist* and *sft* loss narrows with the inclusion of more data, DIST$^2$Loss maintains an edge in generalization. This advantage is further highlighted in the more challenging L2 test protocol, where enhanced coordinate calibration by DIST$^2$Loss significantly improves generalizability to complex tasks.

## 3.4 Textual: Generative Reward Modeling

We apply DIST$^2$Loss to *generative reward modeling* in the RLHF (Reinforcement Learning from Human Feedback) framework, where language models learn from human preference signals. Unlike

---
[2]`liuhaotian/llava-v1.5-7b`

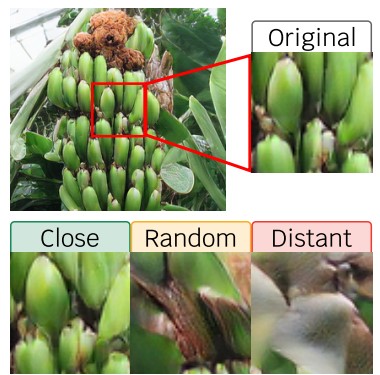 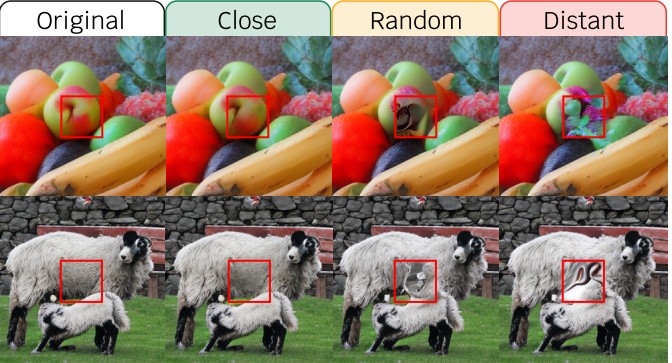

Figure 4: Illustration of token distance effects on image semantics. Each row shows VQ-encoded images with four central tokens replaced by: the original, a nearby token (top-10), a random token, and a distant token (bottom-10). Nearby tokens preserve semantics; random or distant ones cause distortions or semantic shifts.

| Models | Epoch #Params | 50 | | Full (300) | |
|---|---|---|---|---|---|
| | | FID ↓ | IS ↑ | FID ↓ | IS ↑ |
| GigaGAN (Kang et al., 2023) | 569M | - | - | 3.45 | 225.5 |
| LDM-4 (Rombach et al., 2022) | 400M | - | - | 3.60 | 247.7 |
| VQGAN (Esser et al., 2021) | 227M | - | - | 18.65 | 80.4 |
| VQGAN (Esser et al., 2021) | 1.4B | - | - | 15.78 | 74.3 |
| LlamaGen (Sun et al., 2024)-*sft* | 111M | 10.03 | 116.37 | 6.44 | 157.17 |
| LlamaGen-*dist* | 111M | 9.41 | 127.44 | 6.27 | 164.32 |
| LlamaGen (Sun et al., 2024)-*sft* | 343M | 4.24 | 206.74 | 3.08 | 256.07 |
| LlamaGen-*dist* | 343M | **4.18** | **209.41** | **3.04** | **258.19** |

Table 4: Image generation results on ImageNet (Deng et al., 2009) (section D.3)

| Ablation | MAE ↓ | | RMSE ↓ | |
|---|---|---|---|---|
| | mean | std | mean | std |
| Llama-*dist* | **0.092** | **0.017** | **0.124** | **0.026** |
| - Place value weighting | 0.098 | 0.016 | 0.137 | 0.032 |
| - Contrastive loss | 0.099 | 0.015 | 0.139 | 0.020 |
| - Distance-aware target | 0.099 | 0.016 | 0.142 | 0.035 |
| Llama-*sft* | 0.113 | 0.016 | 0.154 | 0.025 |

Table 5: Ablation results on meta linear regression over 10 random seeds.

traditional reward models, generative reward modeling (Zheng et al., 2023; Zhang et al., 2024b) uses next-token prediction within natural language templates, avoiding architectural changes required by classification approaches.

**Setup.** Following prior work (Wang et al., 2024b) on generative reward modeling, we train language models to predict human feedback scores for instruction-response pairs by estimating the sum of the multi-facet scores over the defined range (see section D.2). As a baseline, we train a standard binary classifier. Evaluation is conducted on RewardBench (Lambert et al., 2024) and MT-Bench (Zheng et al., 2023), alongside leaderboard models including UltraRM (Cui et al., 2024), Tulu-v2.5-RM (Ivison et al., 2024), GPT-3.5 (Brown et al., 2020), Claude-3-Haiku (Anthropic, 2024), and Prometheus-2 (Kim et al., 2024b). Open-source model sizes are matched to our backbone LLM[3] (AI@Meta, 2024) for fair comparison.

**Results.** Table 3 summarizes our reward modeling results. DIST[2]Loss shows substantial improvement over the standard cross-entropy loss (*dist* vs. *sft*), highlighting its effectiveness in generative reward modeling. Moreover, generative reward modeling variants outperform the sequential classification baseline (*binary*), suggesting that generative reward modeling is a competitive approach, especially in data-scarce settings, as it fully leverages the pretrained language modeling strengths of the LLM backbone better. The performance gain of *dist* over *binary* is consistent, with notable improvements observed in the Chat Hard and Safety domains.

## 3.5 HIGH-DIMENSION: IMAGE GENERATION

**Effects of Token Distance on Image Semantics** Before training the image generator, we assess how token distance affects image semantics by encoding images, replacing four central tokens, and reconstructing them. Replacements use: (1) top-10 nearest tokens (excluding the original), (2) a random token, and (3) bottom-10 distant tokens. Tokens closely aligned with the original typically

---

[3]meta-llama/Llama-3.1-8B-Instruct

retain the semantic integrity of the image, whereas random replacements cause visual distortions, and more distant tokens introduce new, unrelated concepts, as shown in the reconstructed images in fig. 4. These findings highlight the strong influence of token distances on image semantics.

**Setup.** Following the LlamaGen (Sun et al., 2024) pipeline, we extract discrete features using a pretrained 16×16 compression VQ model and train an autoregressive transformer on the resulting quantized tokens. We adopt mean squared error (MSE) as the distance metric, applied in the embedding space using the VQ model's token embeddings. Inference uses a guidance scale of 2.0, consistent with the original setup.

**Results.** The results in table 4 demonstrate that LlamaGen trained with DIST$^2$Loss consistently outperforms the standard *sft* baseline across various model sizes. This performance advantage is observed at both early (50 epochs) and later (300 epochs) stages of training.

### 3.6 ABLATION STUDY

We further investigate the contribution of each design choice in DIST$^2$Loss using the meta linear regression experiment detailed in section 3.1. Three additional baselines are incorporated by independently ablating each component. First, we examine the impact of ablating *Place value weighting* or *Contrastive loss* for multi-token distances, as described in section A.2. We also assess the effect of substituting the *Distance-aware target* with a label smoothing baseline (Szegedy et al., 2016) of 0.1. Each value is tokenized to three decimal places (e.g., 0.123).

**Results** As shown in table 5, each component contributes positively to DIST$^2$Loss's performance. Notably, the label smoothing baseline, lacking a *distance-aware target*, falls behind the *dist* model by a wide margin. This outcome reinforces our hypothesis that modeling distance relationships is central to DIST$^2$Loss's performance gains.

## 4 RELATED WORK

**Distance Modeling in Discrete Autoregressive Models.** Extensions of LLMs are increasingly adapted for tasks that require precise spatial, temporal, and relational distance modeling. Vision-centric tasks like object detection and segmentation, which rely on generating spatial coordinates, are now addressed by multimodal LLMs (Deitke et al., 2024; You et al., 2024; Zhang et al., 2024a; Xiao et al., 2024). In LLM alignment, generative reward models mimic human feedback to guide instruction tuning (Zheng et al., 2023; Zhang et al., 2024b). LLMs have also been applied in arithmetics (Yuan et al., 2023) and timeseries forecasting (Gruver et al., 2024; Jin et al., 2023), where encoding relational and temporal proximities reduces predictive errors. Additionally, LLMs have shown potential as function regressors (Vacareanu et al., 2024; Song et al., 2024). In robotics, tasks such as manipulation and navigation represent action outputs explicitly through coordinates and joint rotations (Jiang et al., 2023; Brohan et al., 2023; Li et al., 2024) or implicitly via discrete embeddings (Metz et al., 2017; Shafiullah et al., 2022). LLMs are further adapted to fields like geospatial analysis (Manvi et al., 2024), RNA structure prediction (Zablocki et al., 2024), and clinical outcome forecasting (Zheng et al., 2024), where modeling distance relations is essential for understanding spatial and relational data. We introduce a simple and general training objective for distance modeling in LLM-like architectures, broadly applicable across these diverse domains.

**Discretizing Continuous Distribution.** The discretization of continuous distributions is a well-studied area in statistics (Chakraborty, 2015). Discrete analogues of continuous distributions, such as the Laplace (Ghosh et al., 2009) and Gaussian (Canonne et al., 2020), are commonly employed in differential privacy for efficient sampling, often in conjunction with federated learning (Kairouz et al., 2021). For non-analytic continuous distributions, discrete approximations using vector quantization (Van Den Oord et al., 2017) and the Gumbel-Softmax trick (Jang et al., 2022) are common, enabling categorical representations suitable for multimodal generation tasks such as image, video, and audio synthesis (Esser et al., 2021; Yu et al., 2024; Dhariwal et al., 2020). Recently, these quantization approaches have been adopted by general-purpose multimodal generative LLMs (Ge et al., 2024; Wang et al., 2024a; Team, 2024). Building on these methods, we propose a training objective that embeds distance semantics into discrete autoregressive generation.

**Distance Modeling in Loss Functions.** Metric-based objectives have shown effectiveness across applications, such as enhancing explainability in image classification (Choi et al., 2020) and boosting accuracy in few-shot learning (Gao et al., 2022). Likewise, Earth Mover Distance Optimization (EMO) better aligns distributions in language modeling compared to traditional cross-entropy (Ren et al., 2024). More broadly, existing approaches to incorporating distance information into training can be grouped into three categories. *Loss-based* methods modify the objective to account for inter-label distances: ordinal label distribution learning (Wen et al., 2023) models distances between ordinal labels explicitly, and ordinal log-loss (Castagnos et al., 2022) weights the cross-entropy term by label proximity. These formulations are task-specific and not readily compatible with general-purpose LLMs. *Module-based* methods introduce architectural components, such as using distance or graph relationships as priors for transformer attention (Le et al., 2023). In contrast, $DIST^2Loss$ operates solely in the output space and is therefore independent of model architecture. *Target-based* methods redistribute probability mass in the target distribution according to similarity. Class-similarity label smoothing (Chihuang & JaJa, 2021) assigns probability proportional to semantic similarity, and instance-based label smoothing (Maher & Kull, 2021) uses a teacher model's output structure. $DIST^2Loss$ differs in two respects: it operates over the vocabulary of multimodal LLMs (rather than image classification labels), and it derives targets from the task's intrinsic metric without requiring a teacher model. Although $DIST^2Loss$ shares the use of soft targets and KL divergence with knowledge distillation (KD) (Hinton et al., 2015), the two differ fundamentally: KD transfers a teacher model's learned distribution to a student, whereas $DIST^2Loss$ constructs targets deterministically from the ground-truth label and the task's intrinsic metric, requiring no teacher, no auxiliary model, and no additional supervision.

## 5 CONCLUSION

We presented $DIST^2Loss$, a distance-aware objective for discrete autoregressive models. By replacing one-hot targets with reward-weighted distributions derived from token metrics, $DIST^2Loss$ offers a closed-form alternative to reinforcement learning. Experiments across visual grounding, robotic manipulation, reward modeling, and image generation show improved data efficiency, demonstrating the value of distance-aware supervision whenever tokens admit a meaningful metric. We discuss asymptotic behavior under unlimited data in section B. Promising future directions include extending $DIST^2Loss$ to multi-task finetuning settings, generalizing to non-metric structures such as relational or hierarchical outputs, and applying it to additional domains like time-series forecasting.

## ETHICS STATEMENT

$DIST^2Loss$ is a training objective and does not introduce new datasets, human subjects, or sensitive information. All experiments are performed on publicly available datasets and pretrained backbones, with no additional human annotation. As the method modifies only the training loss, ethical considerations inherit from the original datasets and models. Potential concerns such as bias or fairness are therefore bounded by the properties of the underlying backbones and data sources. Since the objective is designed for tasks with numerical or metric outputs, it does not introduce new risks of harmful applications beyond those already present in existing autoregressive models.

## REPRODUCIBILITY STATEMENT

All experiments use publicly available backbones and datasets. Hyperparameter settings, including loss weight and temperature, are documented in Section C. Training follows standard implementations. While training runs incur stochasticity from initialization and data order, we could not conduct extensive statistical analysis across multiple runs due to computational limits. Reported results are therefore based on single or limited runs, but we verify robustness through hyperparameter ablations.

## ACKNOWLEDGEMENTS

This work was partly supported by an Institute of Information & communications Technology Planning & Evaluation (IITP) grant funded by the Korean Government (MSIT) (No.RS-2020-II201361,

Artificial Intelligence Graduate School Program (Yonsei University), No.RS-2022-II220113, Developing a Sustainable Collaborative Multi-modal Lifelong Learning Framework, No.RS-2021-II211343, Artificial Intelligence Graduate School Program (Seoul National University), No.RS-2025- 02263598, Development of Self-Evolving Embodied AGI Platform Technology through Real-World Experience) and the National Research Foundation of Korea (NRF) grant funded by the Korea government (MSIT) (RS-2024-00354218).

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

## A    MULTI-TOKEN DISTANCE

This section provides additional details on the treatment of multi-token distances, clarifying the limitations of $DIST^2Loss$ in this setting and outlining possible workarounds.

### A.1    CREDIT ASSIGNMENT PROBLEM

Extending $DIST^2Loss$ to multi-token sequences implicitly assumes that a global reward can be decomposed into independent token-wise contributions, or that each token can be evaluated while holding the others fixed. In practice, this assumption fails: the model receives undifferentiated feedback across all tokens, without information about which position is responsible for the error. This is the classic *credit assignment problem*. As a result, gradients become noisy and poorly aligned with the true error source, which weakens the learning signal, slows training, and can lead to suboptimal or unstable optimization. It also undermines one of the advantages of $DIST^2Loss$, interpretability, since the induced soft targets no longer reflect a coherent metric over the token space.

For these reasons, we do not apply $DIST^2Loss$ to structured multi-token objectives in our experiments. Instead, we focus on tasks where the reward function decomposes naturally at the token level, such as integers, coordinates, or continuous tokens. These settings preserve the benefits of $DIST^2Loss$: stable training, interpretable reward alignment, and computational efficiency. Generalizing to multi-token objectives would require explicit credit assignment mechanisms or structured training methods, which we view as an important direction for future work.

### A.2    WORKAROUNDS

Consider a multi-token case where each element in the metric space consists of a sequence of tokens from the vocabulary, denoted $x_{\text{multi}} \in \mathcal{X}_{\text{multi}}$ with $x_{\text{multi}} = (x_1, \ldots, x_L)$. For instance, this could represent a multi-digit integer split into individual tokens. Applying multi-token objectives directly in autoregressive models trained with teacher-forcing is extremely inefficient, as it requires training-time sequence generation. To circumvent this limitation, we propose two practical alternatives.

**Contrastive Target Augmentation**    Instead of evaluating all possible multi-token sequences, we propose sampling a contrastive multi-token candidate $\bar{x} \in \mathcal{X}_{\text{multi}}$ for training. Such a candidate is selected from nearby neighbors of the target $x$ in the metric space, without reference to the training model $f_\theta$. For example, in the case of integer sequences, 39 might be chosen as a close neighbor to the target 40, with each digit tokenized separately.

For each token in the sequence, we extend the target distribution by incorporating the negative sample $\bar{x}$. The contribution of each token in $\bar{x}$ to the overall distance is defined based on its position-wise difference from the target $x$. For instance, when the target is 40, the negative sample 39 is assigned a token-wise distance where the tens digit 3 has distance 0 from the target's 4, while the units digit 9 has a distance of 1 from the target's 0. We then concatenate the logits of $x$ and the selected logit $\bar{x}$ at each token position, forming an extended likelihood distribution. The distance loss $\mathcal{L}_{\text{dist}}$ is applied to this extended distribution.

**Place Value Weighting**    For tasks involving multi-digit integers or sequences where token positions have different significance, we introduce place value weighting. In this approach, tokens are weighted according to their positional importance, so that differences in higher place values have a greater impact on the loss. For example, in a multi-digit integer setting, we directly multiply the distance loss by the place value weight for each token, assigning more weight to tokens in higher positions. Let $x_{\text{multi}} = (x_1, \ldots, x_L)$ represent the target sequence, with $x_i$ denoting the digit in the $i$-th place (e.g., thousands, hundreds, tens, units). The place-weighted loss is formulated as:

$$\mathcal{L}_{\text{place}} = \sum_{i=1}^{L} w_i \cdot \mathcal{L}_{\text{dist}}(x_i) \tag{9}$$

where $w_i$ is the place weight for position $i$: 4 (thousands), 3 (hundreds), 2 (tens), and 1 (units).

| Coefficient ($\alpha$) | Accuracy (%) |
|---|---|
| 1.0 | 77.3 |
| 0.2 | 77.8 |
| 0.1 | 85.3 |
| 0.02 | 80.7 |
| 0.01 | 79.9 |
| 0.005 | 76.7 |
| 0.001 | 75.6 |
| SFT | 75.4 |

Table 6: **Hyperparameter sensitivity analysis** on the loss weight coefficient $\alpha$. Results shown for reward modeling.

## B  DISCUSSION

### B.1  ASYMPTOTIC BEHAVIOR

A potential concern is that the advantages of DIST$^2$Loss diminish with unlimited data and compute. While true in theory, this setting is not representative of practical training. Realistic applications operate with limited supervision, moderate model capacity, and constrained compute, where inductive biases are critical. Under these conditions, DIST$^2$Loss provides consistent improvements with negligible overhead, as demonstrated in all reported experiments using standard backbones, realistic dataset sizes, and established evaluation protocols.

## C  ADDITIONAL EXPERIMENTS

### C.1  HYPERPARAMETER SENSITIVITY

DIST$^2$Loss introduces two tunable hyperparameters.

**Loss weight $\alpha$.** We fix $\alpha = 0.1$ unless otherwise noted. A sweep in reward modeling shows robustness across a wide range; performance drops only when $\alpha$ becomes too small, effectively reducing the method to SFT. We additionally conduct sensitivity analysis on table 6, which confirms that DIST$^2$Loss improves over the base SFT for a wide range of $\alpha$.

**Temperature $\tau$.** Controls the sharpness of the soft target distribution. We derive $\tau$ from the structure of the token space rather than tuning it.

For *decimal digit tokens* ($\mathcal{V}_d = \{0, \ldots, 9\}$), the squared Euclidean distance on the integer lattice gives $d(v, x_t) = (v - x_t)^2$. Setting $\tau = 1$ yields a likelihood kernel $\exp(-(v - x_t)^2)$, which corresponds to a discretized unit-variance Gaussian centered at $x_t$. This is a natural default that assigns geometrically decaying probability to tokens farther from the target.

For *VQ codebook tokens*, similarity scales are not fixed a priori: identical score gaps can correspond to different semantic distances depending on codebook training. We therefore select $\tau$ using an information-theoretic criterion. A uniform distribution over a vocabulary of size $K$ has entropy $\log K$. We choose $\tau$ so that the induced soft target has comparable entropy, ensuring consistent sharpness across different vocabulary sizes. For a VQ codebook with $K = 16{,}384$, this yields $\tau \approx 9.7$.

Place value weights in multi-digit numbers are fixed by construction and not tunable.

### C.2  ROBUSTNESS TO TASK AND METRIC VARIATIONS

To assess generalization, we conducted two experiments.

**Task generalization.** We evaluated whether DIST$^2$Loss fine-tuning impairs unrelated tasks. A reward-modeling model tested on MMLU and a visual grounding model (RefCOCO) tested on RealWorldQA both show negligible degradation, as reported in table 7.

| Model | Reward Accuracy (%) | MMLU Accuracy (%) |
|---|---|---|
| Backbone (Llama-3.1-8B) | - | 44.5 |
| SFT | 75.3 | 42.8 |
| DIST$^2$Loss | 85.3 | 43.9 |

| Model | RealWorldQA Accuracy (%) |
|---|---|
| Backbone (Phi-3.5V) | 54.4 |
| DIST$^2$Loss on RefCOCO | 54.3 |

Table 7: **Catastrophic forgetting analysis.** Top: fine-tuning with DIST$^2$Loss for reward modeling yields minimal degradation on the general-purpose MMLU benchmark. Bottom: fine-tuning with DIST$^2$Loss for visual grounding (RefCOCO) preserves performance on the out-of-domain Real-WorldQA visual understanding benchmark.

| Metric | Accuracy (%) |
|---|---|
| DIST$^2$Loss (Euclidean) | 85.3 |
| DIST$^2$Loss (Random) | 76.0 |
| SFT | 75.3 |

Table 8: **Sanity check with a contradictory metric.** Using a *random* distance metric provides no improvement over SFT, confirming that the semantic validity of the metric is essential for DIST$^2$Loss. Results shown for reward modeling.

**Contradictory metric.** We trained reward models with randomly assigned distances between labels. As shown in table 8, this yielded no gains over SFT, confirming that improvements arise when distances capture meaningful structure.

# D  IMPLEMENTATION DETAILS

## D.1  GLOBAL SETUPS

We use the HuggingFace Trainer (Wolf et al., 2020) and TRL trainer (von Werra et al., 2020) with DeepSpeed ZeRO-3 (Ren et al., 2021) and the AdamW optimizer (Loshchilov & Hutter, 2019). The base foundational models are detailed in table 9, with computational requirements specified in table 10.

## D.2  TASK-SPECIFIC SETUPS

**Toy: Learning to Regress** The learning rate is set to $2e^{-5}$ with a linear decay schedule and no warmup. Training epochs are configured to ensure each model is exposed to approximately 250 samples to prevent underfitting. For example, with a training dataset size of 2, the epoch count is set to 125. Each experiment is repeated five times with random seeds $[1:5]$ for statistical stability.

**Textual: Generative Reward Modeling** For fine-tuning, the helpsteer2 dataset (Wang et al., 2024b) was reformatted into an instruction-following structure, where scores for each of the five categories were designated as model outputs. The model was trained for two epochs with a learning rate of $1 \times 10^{-5}$ using the paged Adam optimizer (Kingma & Ba, 2015). The prompt used during training is illustrated in fig. 5. During inference, a logit-based score prediction function was implemented to evaluate two samples by generating score probabilities on a 0-20 points scale. The model calculated weighted averages from the softmax probabilities, assigning a final reward based on higher scores for preferred outputs.

> **Prompt for Generative Reward Evaluation**
>
> Please act as an impartial judge and evaluate the quality of the response provided by AI assistant to the user question displayed below. Your evaluation should consider five factors helpfulness, correctness, coherence, complexity, verbosity. Here's brief explanation of each factor:
> - Helpfulness: Overall helpfulness of the response to the prompt.
> - Correctness: Inclusion of all pertinent facts without errors.
> - Coherence: Consistency and clarity of expression.
> - Complexity: Intellectual depth required to write response (i.e. whether the response can be written by anyone with basic language competency or requires deep domain expertise).
> - Verbosity: Amount of detail included in the response, relative to what is asked for in the prompt.
> Do not allow the length of the responses to influence your evaluation. Be as objective as possible. Please first provide an overall score over model response. You must provide overall score as a number between 0 and 20.
> Then provide a set of 5 score over model response. Only provide the score as a number between 0 and 4.
> [User Question]
> {*user input*}
> [Start of Model Response]
> {*model response to evaluate*}
> [End of Model Response]

Figure 5: Instruction-tuning prompt template for generative reward modeling.

**Multimodal: Visual Grounding** For fine-tuning, we concatenate the training sets of RefCOCO, RefCOCO+, and RefCOCOg (Kazemzadeh et al., 2014; Mao et al., 2016; Yu et al., 2016). All images are resized to $1024 \times 1024$ to constrain the range of generated digits, with coordinate values rounded to the nearest integer. During inference, outputs that cannot be parsed as bounding box coordinates are considered incorrect. Training is conducted with a learning rate of $2e^{-5}$, 100 steps of linear warmup, and a total of three epochs.

**Embodied: Robotic Manipulation** We convert the VIMA dataset (Jiang et al., 2023) into an instruction-tuning-compatible format using the provided script from the LLaRA (Li et al., 2024) repository . The pretrained LLaVA-1.5 (Liu et al., 2024) model is then fine-tuned on the object manipulation task. Following (Li et al., 2024), we incorporate auxiliary objective augmentations from the same repository into the training set. We the oracle object detection labels for evaluation. Training is conducted with a learning rate of $2e^{-5}$, using a 0.3 ratio of linear warmup and cosine decay over two epochs.

**High-Dimension: Image Generation** We employ the pretrained image vector quantization model from the LlamaGen (Sun et al., 2024) repository . All images are resized to $384 \times 384$ using random center cropping. During evaluation, images are generated at $384 \times 384$ and then resized to $256 \times 256$ for model-based metric computations. Classifier-free guidance with a scale of 2.0 is applied during inference. Experimental protocols strictly adhere to the repository's guidelines.

## D.3 BASELINE SCORES

**Embodied: Robotic Manipulation** For LLaRA$_{\text{sft}}$, we adopt results from Tables 15, 17, and 19 of the original paper (Li et al., 2024), using D-inBC + Aux with all six auxiliary tasks (epoch: 2, iteration: 14) for data sizes of 0.8k, 8k, and 80k. Notably, at the 80k scale, using all auxiliary tasks does not outperform using only a subset, as reported in Table 1 of the same paper. However, we adopt the former for consistency and generalizability across different scales.

| Experiment Type | Size | Backbone |
|---|---|---|
| Toy (3.1) | 1B | `meta-llama/Llama-3.2-1B-Instruct` (AI@Meta, 2024) |
| Textual (3.4) | 8B | `meta-llama/Llama-3.1-8B-Instruct` (AI@Meta, 2024) |
| Multimodal (3.2) | 3.8B | `microsoft/Phi-3-mini-4k-instruct` (Abdin et al., 2024) |
| Embodied (3.3) | 7B | `liuhaotian/llava-v1.5-7b` (Liu et al., 2024) |
| High-Dimension (3.5) | 343M | Scratch (Sun et al., 2024) |

Table 9: Base foundational models used for finetuning in each experiment type.

| Experiment Type | GPU Model | VRAM (GB) | # GPUs |
|---|---|---|---|
| Toy (3.1) | RTX 3090 | 24 | 1 |
| Textual (3.4) | A6000 | 48 | 4 |
| Multimodal (3.2) | A6000 | 48 | 4 |
| Embodied (3.3) | L40S | 48 | 8 |
| High-Dimension (3.5) | L40S | 48 | 8 |

Table 10: Computational requirements for each experiment are reported per single run; multiple runs may be needed depending on configuration or random seeds.

| Models | MAE↓ mean | std | RMSE↓ mean | std | MAE↓ mean | std | RMSE↓ mean | std | MAE↓ mean | std | RMSE↓ mean | std | MAE↓ mean | std | RMSE↓ mean | std | MAE↓ mean | std | RMSE↓ mean | std |
|---|---|---|---|---|---|---|---|---|---|---|---|---|---|---|---|---|---|---|---|---|
| | Training Problems: 1 | | | | Training Problems: 2 | | | | Training Problems: 3 | | | | Training Problems: 4 | | | | Training Problems: 5 | | | |
| Llama-3.2 1B (AI@Meta, 2024)-*sft* | **0.2** | 0.039 | **0.243** | 0.05 | 0.182 | 0.014 | 0.221 | 0.02 | 0.17 | 0.039 | 0.216 | 0.056 | 0.152 | 0.018 | 0.2 | 0.035 | 0.159 | 0.03 | 0.211 | 0.042 |
| Llama-3.2 1B-*vocab* | **0.2** | 0.039 | **0.243** | 0.05 | **0.157** | 0.022 | **0.202** | 0.023 | 0.167 | 0.031 | 0.215 | 0.05 | 0.127 | 0.007 | 0.166 | 0.016 | 0.133 | 0.006 | 0.172 | 0.009 |
| Llama-3.2 1B-*dist* | 0.21 | 0.032 | 0.259 | 0.037 | 0.17 | 0.006 | 0.212 | 0.008 | **0.147** | 0.032 | **0.198** | 0.035 | **0.114** | 0.017 | **0.146** | 0.022 | **0.122** | 0.023 | **0.165** | 0.032 |
| | Training Problems: 6 | | | | Training Problems: 7 | | | | Training Problems: 8 | | | | Training Problems: 9 | | | | Training Problems: 10 | | | |
| Llama-3.2 1B-*sft* | 0.144 | 0.024 | 0.185 | 0.032 | 0.129 | 0.011 | 0.173 | 0.02 | 0.119 | 0.017 | 0.154 | 0.019 | 0.115 | 0.012 | **0.154** | 0.022 | 0.113 | 0.016 | 0.154 | 0.025 |
| Llama-3.2 1B-*vocab* | 0.122 | 0.016 | 0.162 | 0.017 | 0.141 | 0.034 | 0.189 | 0.046 | 0.122 | 0.029 | 0.164 | 0.035 | 0.122 | 0.014 | 0.163 | 0.023 | 0.111 | 0.008 | 0.151 | 0.014 |
| Llama-3.2 1B-*dist* | **0.113** | 0.018 | **0.153** | 0.021 | **0.104** | 0.013 | **0.148** | 0.031 | **0.104** | 0.035 | **0.15** | 0.081 | **0.112** | 0.053 | 0.163 | 0.093 | **0.092** | 0.017 | **0.124** | 0.026 |

Table 11: Meta linear regression experiment results on one to ten training problems and 1,000 test problems, with scores averaged over five random seeds.

| Models | Model Type | Average | AlpacaEval | | | Do-Not-Answer | HumanEvalPack | | | | | | - |
|---|---|---|---|---|---|---|---|---|---|---|---|---|---|
| | | | Easy | Hard | Length | | CPP | GO | Java | Javascript | Python | Rust | |
| Llama-3.1-8B (AI@Meta, 2024)-*binary* | Seq. Classifier | 58 | 94.5 | 94.7 | 76.3 | 16.9 | 54.9 | 55.8 | 56.1 | 52.4 | 48.5 | 56.7 | - |
| Llama-3.1-8B-*sft* | Generative | 75.3 | 89.0 | 97.9 | 77.9 | 44.9 | 84.1 | 80.5 | | 83.5 | | 81.1 | - |
| Llama-3.1-8B-*dist* | Generative | 85.3 | 97.0 | 98.9 | 88.4 | 78.7 | 89.6 | 90.2 | 89.6 | 87.8 | 90.2 | 85.4 | - |

| | LLMBar | | | | | MATH | MT-Bench | | | Refusal | | XSTest | |
|---|---|---|---|---|---|---|---|---|---|---|---|---|---|
| | Adver. GPTInst | Adver. GPTOut | Adver. Manual | Adver. Neighbor | Natural | PRM | Easy | Hard | Medium | Dangerous | Offensive | Should Refuse | Should Respond |
| Llama-3.1-8B-*binary* | 13.6 | 36.2 | 23.9 | 24.6 | 61.5 | 92.8 | 64.3 | 62.1 | 62.5 | 0.4 | 0.3 | 22.7 | 92.0 |
| Llama-3.1-8B-*sft* | 32.6 | 63.8 | 32.6 | 29.1 | 82.0 | 84.1 | 96.4 | 78.3 | 90.0 | 93.0 | 99.0 | 92.9 | 76.0 |
| Llama-3.1-8B-*dist* | 57.6 | 72.3 | 67.4 | 63.4 | 84.0 | 84.1 | 100.0 | 75.7 | 92.5 | 96.0 | 100 | 94.8 | 88.0 |

Table 12: Fine-grained statistics on model performance on RewardBench (Lambert et al., 2024).

**High-Dimension: Image Generation** We use the class-conditional ImageNet 256×256 results with CFG 2.0 from Table 9 of the LlamaGen paper (Sun et al., 2024) as baselines.

# E   EXTENDED QUANTITATIVE RESULTS

**Toy: Learning to Regress** We provide scores corresponding to fig. 3 in the main paper in table 11.

**Textual Task: Generative Reward Modeling** Detailed results for each data source in Reward-Bench (Lambert et al., 2024) are reported in table 12.

# F   ADDITIONAL QUALITATIVE SAMPLES

**Textual: Generative Reward Modeling** Figure 7 shows inference results of Llama-based generative reward model trained with DIST$^2$Loss.

**Multimodal: Visual Grounding** Figure 8 presents qualitative results from visual grounding experiments, comparing the base cross-entropy loss with our proposed DIST$^2$Loss. To further assess metric sensitivity, we measure the mean IoU on RefCOCO testA restricted to cases where both models predict the wrong object (IoU < 0.5). Correct predictions saturate IoU, so shared failures better reveal how closely each model preserves geometric structure. As shown in table 13, DIST$^2$Loss produces predictions that are consistently closer to the ground truth even when incorrect.

| Model | IoU (RefCOCO testA, hard cases) |
|---|---|
| SFT | 31.0 |
| DIST$^2$Loss | 40.3 |

Table 13: **Metric sensitivity on hard cases.** Mean IoU on RefCOCO testA examples where both models predict the wrong bounding box (IoU < 0.5). DIST$^2$Loss predictions remain closer to the ground truth, indicating improved geometric alignment.

---

**Instruction-Response Pair Sample for Object Manipulation**

[Prompt]
<image0> <task>Sweep any <p>polka dot small block</p> at (0.500, 0.617), 0.203, 0.578 into <p>blue three-sided rectangle</p> at (0.500, 0.625), 0.297, 0.398 without exceeding <p>plastic line</p> at (0.500, 0.617), 0.328, 0.047.</task>
Can you explain what needs to be done to perform the task in this scene?
Every action you take must include two locations in the format of (x, y) and one clockwise rotation angle in the format of <r>[r]</r>. The first location is the image coordinate where you start to sweep the object using a spatula, and the second location is where you stop sweeping. The image coordinate ranges from 0 to 1. The rotation angle indicates how many degrees you rotate the spatula clockwise, and it ranges from -359 to 359.

[Answer]
Step 1: Sweep the <p>polka dot small block</p> at (**0.629**, **0.805**), rotate <r>[**0**]</r> degrees, and stop at (**0.500**, **0.609**).

Figure 6: Example of an instruction-response pair from the VIMA dataset Jiang et al. (2023), illustrating an object manipulation task. Metric-related output tokens are bolded for emphasis.

**Query**: Who created the Superman cartoon character?

**"Chosen" Answer**: Superman, the iconic comic book superhero, was created by writer Jerry Siegel and artist Joe Shuster. Superman first appeared in Action Comics #1, which was published by Detective Comics, Inc. (later DC Comics) in June 1938. The character's immense popularity established him as one of the most enduring and recognizable figures in the superhero genre.

**DIST$^2$Loss** helpfulness score expectation: 2.81

**"rejected" Answer**: The Superman cartoon character was created by Jerry Siegel and Joe Shuster in 1938.

**DIST$^2$Loss** helpfulness score expectatione: 0.96

**Query**: Hi, I'm in the mood for a Bloody Mary. Can you give me a recipe for making one?

**"Chosen" Answer**: Certainly! The Bloody Mary is a classic cocktail known for its bold flavor and is commonly enjoyed as a brunch drink. Here's a basic recipe for making a single serving:
### Ingredients:
- 1 1/2 ounces (45 ml) vodka
- 3 ounces (90 ml) tomato juice
- 1/2 ounce (15 ml) lemon juice, freshly squeezed
- 1 dash of Worcestershire sauce
- Tabasco sauce (to taste)
- Ground black pepper (to taste)
- Celery salt (to taste)
- Horseradish (optional, to taste)
- 1 celery stalk (for garnish)
- 1 lemon wedge or lime wedge (for garnish)
- Ice cubes

### Instructions:
1. **Prepare the Glass:**
- Rub the lemon or lime wedge along the lip of a tall glass (often a highball glass).
- Dip the wetted rim into celery salt if you like a salted rim.
2. **Mix the Drink:**
- Fill the glass with ice cubes.
- Add vodka, tomato juice, and lemon juice to the glass.
- Add Worcestershire sauce, a few dashes of Tabasco, a pinch of celery salt, ground black pepper, and horseradish (if using).
- Stir well to combine all the ingredients.
3. **Garnish:**
- Garnish with a celery stalk and a lemon or lime wedge on the rim.
4. **Adjust and Serve:**
- Taste and adjust the seasoning, adding more Tabasco, lemon juice, or other ingredients as necessary.
- Serve immediately.

### Variations and Tips:
- **Garnish Variations:** You can also garnish with olives, pickles, cocktail onions, or even a strip of crispy bacon.
- **Spice Level:** Adjust the spiciness to your liking with more or less Tabasco or by adding hot sauce of your choice.
- **Clamato:** Some people like to use Clamato (a mixture of clam and tomato juices) instead of plain tomato juice for a different twist known as a Bloody Caesar.
- **Make it a Virgin:** To make a non-alcoholic version, simply omit the vodka for a Virgin Mary.
Feel free to customize your Bloody Mary to your taste preferences. Enjoy your cocktail!

**DIST$^2$Loss** Overall score expectation: 1.33

**"rejected" Answer**: Sure, here's a quick and easy recipe for making a delicious Bloody Mary:
Ingredients:
- 2 ounces vodka
- 3-4 dashes Worcestershire sauce
- 1-2 teaspoons hot sauce
- 1 lime wedge for garnish
- 2 ounces tomato juice
- 1 ounce horseradish
- 1/2 ounce beet juice
- 1/2 ounce olive juice
Instructions:
Fill an Old-Fashioned glass with ice cubes. In a shaker, combine the vodka, Worcestershire sauce, hot sauce, lime wedge, and tomato juice. Shake well to combine. Strain the mixture into the prepared glass and top off with beet juice and olive juice. Garnish with a lime wedge and serve.

**DIST$^2$Loss** Overall score expectation: 1.20

Figure 7: Qualitative examples from the generative reward modeling experiment.

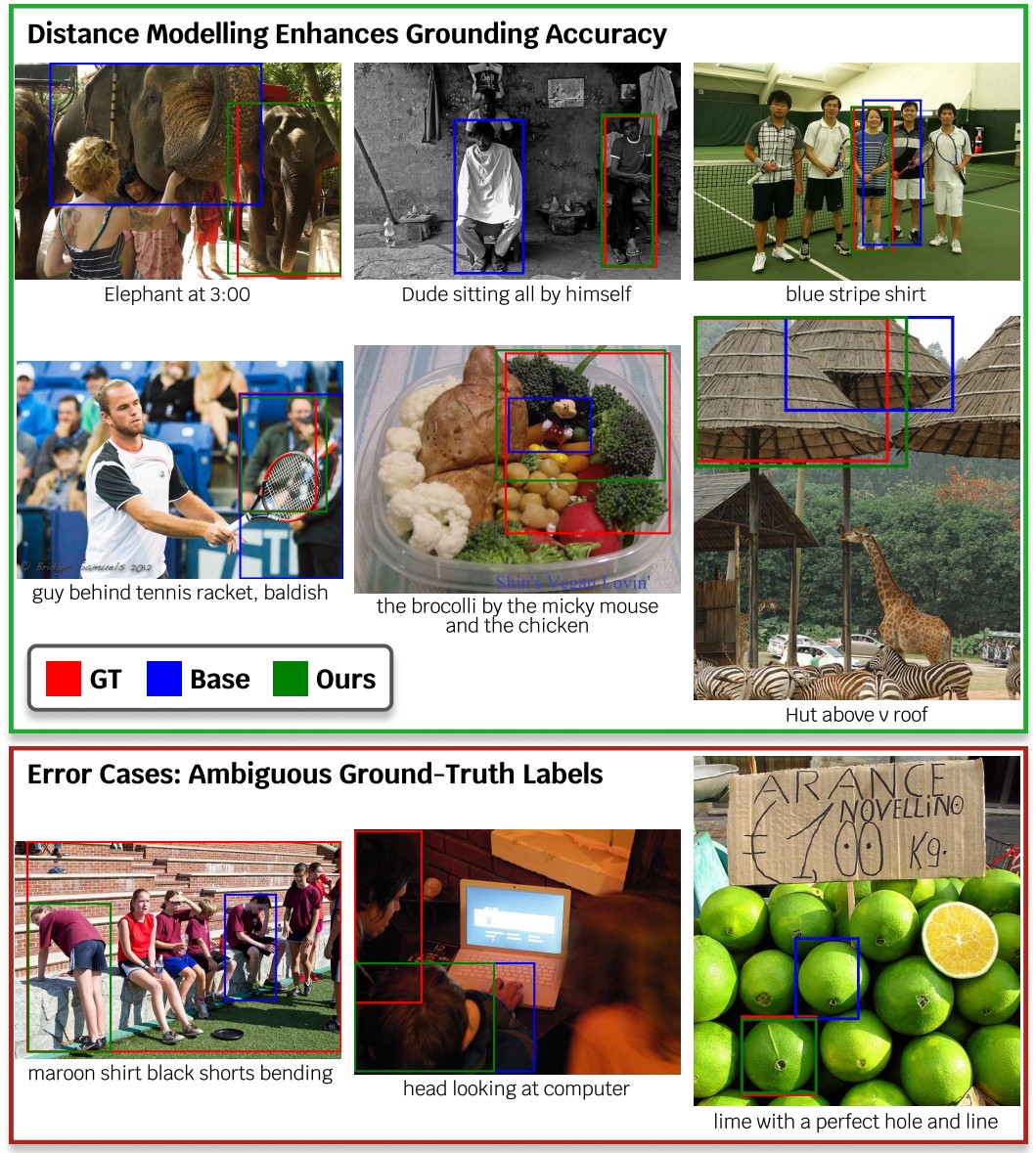

Figure 8: Qualitative examples from the visual grounding experiment. Top: our proposed DIST$^2$Loss loss demonstrates higher visual grounding accuracy compared to the standard cross-entropy loss. Bottom: A manual examination of inference results reveals that a substantial portion of the RefCOCO dataset (Kazemzadeh et al., 2014; Mao et al., 2016; Yu et al., 2016) contains labels that are ambiguous, even for human annotators, which may lead to underestimation of model performance.

