# OpenReview forum: "Teaching Metric Distance to Discrete Autoregressive Language Models"
_ICLR.cc/2026/Conference — ICLR 2026 Poster_

### Official Review · Reviewer_Pvxr · 2025-10-27

**Soundness:** 3
**Presentation:** 3
**Contribution:** 3
**Rating:** 4
**Confidence:** 4

**Summary:**

The paper addresses a key limitation in adapting large language models (LLMs) to tasks involving numerical or metrically structured data. Conventional fine-tuning methods treat tokens as categorical variables, ignoring the intrinsic distance relationships among them—such as numerical proximity or spatial metrics. To overcome this, the authors propose DIST² loss, a new objective that integrates predefined distance relationships between tokens into the autoregressive loss used during training. This approach enables LLMs to better capture metric structure while remaining compatible with standard categorical token modeling. Empirical results demonstrate substantial performance improvements across tasks where distances are semantically meaningful, including object detection, object manipulation, reward modeling, and image generation.

**Strengths:**

* The paper is clearly written and easy to follow.
* The problem is important and timely, addressing a fundamental gap in how LLMs handle numerical and metric relationships.
* Empirical results show significant improvements across diverse downstream tasks, including those with limited data
* The proposed method integrates smoothly into existing LLM training objectives, maintaining compatibility with categorical token modeling.

**Weaknesses:**

The paper does not clearly discuss how different types of numerical or metric data (e.g., integers, floats, directions) are handled under the same framework. It remains unclear whether the model improves actual numerical reasoning accuracy, beyond performance metrics on downstream tasks. There is a lack of explicit evaluation of the model’s ability to reason about true numerical distances or relationships, which would more directly validate the method’s intended benefits. See the questions below.

**Questions:**

1. Is the same metric used for every numeral token, or are different metrics applied depending on token type (e.g., integers vs. floats vs. directions)?
2. Are metric-based losses applied to text-only tokens, or only to numerically meaningful ones?
3. Are such tokens, on which the metric/distance-based loss is applied, extracted manually?
4. While the paper shows improvements in downstream tasks, could the authors demonstrate that the model achieves more accurate numerical reasoning (e.g., better alignment between predicted and true distances)? The authors could consider a design of a controlled experiment where digits/numbers are extracted from data and the trained models are asked to evaluate the distances or any related tasks. The results are then compared with true numerical distance values to directly test metric reasoning performance.

The current score reflects the questions above. I will consider updating the rating once my concerns have been resolved.

---

> ### Author Response · Authors · 2025-11-17
>
> We thank the reviewer for highlighting the importance of our problem formulation and for acknowledging the significance of our experimental results. Several concerns arise from ambiguities in our presentation, and we appreciate the opportunity to clarify these points. Our responses are as follows:
>
> ---
>
> ### **Question 1. Response: Each numerical field naturally induces a metric.**
>
> For scalar numerical values (integers, floats), we default to the Euclidean metric (MSE), which is the standard choice in embodied control and regression tasks (e.g., joint-angle prediction in embodied agent [1]). Toy experiment results in Section 3.1 additionally reports results with MAE, showing that DIST$^2$Loss is robust to the choice of metric as long as it reflects the natural geometry of the field.
>
> For vector-quantized representations, we use cosine similarity because this is the metric used during codebook training. More broadly, latent or model-induced representations typically come with a canonical metric determined by the objective used to build the representation.
>
> ---
>
> ### **Question 2. Response: Metric-based loss is applied only to numerically meaningful fields.**
>
> We apply DIST$^2$Loss solely to token positions belonging to metric-structured outputs (e.g., coordinates, angles, VQ indices, scalar values). All other tokens, including natural-language tokens, are trained exclusively with standard cross-entropy. This ensures that the metric supervision affects only the fields whose semantics justify it.
>
> ---
>
> ### **Question 3. Response: Metric-relevant tokens are identified automatically from the vocabulary.**
>
> Identification does not require manual annotation.
> - For integers, we locate all integer tokens in the vocabulary and check for set membership.
> - For floating-point values, we use a greedy span-detection algorithm that accounts for decimal points.
> - For vector-quantized tokens, identification is trivial because the corresponding vocabulary indices form a dedicated range.
>
> ---
>
> ### **Question 4. Response to the reviewer’s concern about numerical reasoning evaluation**
>
> We agree that evaluating how well the model respects metric structure is important. However, evaluating numerical reasoning by extracting digits or numbers from text and directly querying their distances would not measure the phenomenon DIST$^2$Loss is designed to affect. This is not because such an experiment is difficult, but because it would *mischaracterize* what our method modifies.
>
> DIST$^2$Loss is *not* designed to enhance general numerical reasoning or arithmetic. It modifies the autoregressive target distribution **only** for tokens whose distances have *task-specific semantic meaning*, such as discretized spatial coordinates, orientation bins, reward magnitudes, or latent codes. These distances are defined **within each task’s token space**, not as universal numeric values.
>
> Therefore, evaluating the model using isolated numbers outside those contexts would not measure the behavior that DIST²Loss is intended to improve.
>
> More concretely:
>
> - **Metric sensitivity is improved only within the token space where distances are defined.** DIST$^2$Loss does not aim to improve general numerical competence.
> - **Distances are task-specific**, e.g., grid indices for detection/manipulation, discretized rewards, or voxel indices for image generation. “True numerical distance” beyond these task definitions is not meaningful. For example, models trained for 1-5 Likert scale reward modeling would not be able to model integers larger than 5.
> - **Our downstream evaluations already measure domain-intrinsic metric fidelity**, such as spatial alignment or magnitude accuracy, which directly reflect how well predictions respect the metric structure encoded by DIST$^2$Loss.
> - An extrinsic evaluation using free-form numbers would probe abilities outside the causal pathway of the method and would not produce interpretable validation.
>
> **Alternative experiment demonstrating metric behavior**
>
> That said, we agree that the paper can benefit from a more interpretable examination of metric behavior. To assess how well the model captures task-specific geometric relationships, we conduct an additional analysis on the RefCOCO visual grounding task, measuring how closely each model’s predicted bounding box aligns with the ground-truth box.
>
> We focus on the cases where both models predict the wrong object, because correct predictions saturate IoU, while shared failures reveal *how closely each model preserves the underlying geometric structure*. In this setting, DIST$^2$Loss produces predictions that are consistently closer to the ground truth, demonstrating improved metric sensitivity.
>
>
> | model | IoU (RefCOCO testA Hard) |
> |:------------:|:--------|
> | SFT | 31.0 |
> | DIST$^2$Loss | 40.3 |
>
> ---
>
> ### **References**
>
>
> [1] Pi-Zero ([https://arxiv.org/pdf/2410.24164v1](https://arxiv.org/pdf/2410.24164v1))

---

> ### Comment · Reviewer_Pvxr · 2025-11-25
>
> Thank you for the responses. The authors have addressed many of the questions, so I am willing to increase to score to 6. However, my concern for the last question remains and I elaborate on it here in case AC may need take it into account.
>
> I understand that numerical reasoning is not the goal of the method. Then my suggestion is that the authors should be more careful in framing the contributions to avoid misinterpretation.
>
> The reason why I stress this is that it might raise a concern that how a model can perform well on distance-related downstream tasks without true numerical reasoning capability, or whether there is some form of shortcuts that it exploits to succeed in the tasks. A controlled experiment would help clarify this, as well as to see the true effect of the proposed DIST$^2$Loss.
>
> When I say true numerical reasoning, I expect that when the model outputs, say, a line A of length 5 and a line B of length 9, it knows that B is longer than A, or something similar for direction (closer vs. farther). And I understand the evaluation is task-specific, which is what I expected. I also expected a comparison with the true arithmetic values, not to compare between models (which makes no sense to me).

---

### Official Review · Reviewer_7BSi · 2025-10-29

**Soundness:** 3
**Presentation:** 3
**Contribution:** 2
**Rating:** 2
**Confidence:** 5

**Summary:**

This paper propose to learn the connections between discrete outputs, such the correlations between them can be captured during the learning process.

**Strengths:**

1. The reported results are marginally improved compared with existing methods.

2. The source code is produced.

**Weaknesses:**

1. The motivation of learning correlations among discrete concepts/semantics is not novel, which has been largely explored in the knowledge distillation works for a decade.

2. The proposed DIST^2 loss is just a combination of loss used in knowledge distillation and KL-divergence, which is also proposed in 'Distilling Knowledge from Graph Convolutional Networks. CVPR 2020'.

3. Overall, this work lacks of novelty and is with limited technical contribution.

**Questions:**

Please refer to the Weaknesses.

---

> ### Author Response · Authors · 2025-11-17
>
> We thank the reviewer for the comments. The concerns appear to stem from a misunderstanding of the purpose and technical novelty of DIST$^2$Loss. The method is not a variant of knowledge distillation (KD), nor is the contribution the introduction of KL divergence. The key distinctions are clarified below.
>
> ---
>
> ## **1. Core novelty: incorporating metric structure into autoregressive training**
>
> The novelty of DIST$^2$Loss lies in defining a **metric-induced target distribution** for discrete autoregressive models.
> KL is simply the vehicle for enforcing this target; the contribution is the mechanism for constructing a **geometric prior over tokens** based on task-intrinsic metric relationships (e.g., Euclidean distance, angular distance, VQ codebook geometry).
>
> This provides a general framework for adapting LLM-style models to domains where outputs possess meaningful geometric structure; something standard CE training does not capture.
>
> ---
>
> ## **2. DIST$^2$Loss is not knowledge distillation**
>
> Knowledge distillation requires:
>
> - a teacher model,
> - teacher logits or embeddings,
> - transfer of semantic or structural information from teacher to student.
>
> DIST$^2$Loss uses none of these components.
>
> The target distribution is **deterministically derived from the ground-truth value and its metric neighborhood**, not from a teacher. There is no external supervision, no softened teacher logits, and no distillation of semantic correlations.
>
> KD aligns predictions with a teacher distribution.
> DIST$^2$Loss aligns predictions with a **metric-driven geometric prior** that is intrinsic to the task.
>
> The work cited by the reviewer [1] concerns KD for graph convolutional networks, which is unrelated in both goal and formulation.
>
> We will revise the manuscript to make this distinction explicit.
>
> ---
>
> ## **3. Technical contribution**
>
> The contribution is not a simple combination of losses. It is the **formulation and empirical validation of an output-space metric prior** that can be applied in a task-agnostic manner to discrete autoregressive models.
>
> This yields consistent improvements across five distinct domains (regression, grounding, manipulation, reward modeling, and image generation) without additional supervision or architectural changes, a setting in which KD-based methods do not operate.
>
>
> We appreciate the reviewer’s feedback and will strengthen the manuscript to prevent these misunderstandings.
>
> ---
>
>
> ### **References**
>
> [1] Yang, Yiding, et al. "Distilling knowledge from graph convolutional networks." 2020.

---

### Official Review · Reviewer_qCXL · 2025-11-01

**Soundness:** 3
**Presentation:** 3
**Contribution:** 3
**Rating:** 6
**Confidence:** 4

**Summary:**

The work aims to incorporate an additional regularisation term into the loss of an LLM which more explicitly forces the model to learn a representation of distance (in terms of the metric used in the regularisation term). The distance regulariser is also phrased in terms of entropy-regularisation in policy optimisation. Extensive experiments are shown, including an ablation study, which consistently support the utility of the proposed regularisation term.

**Strengths:**

## Originality
The work is fairly original, particularly in the exact way they approach incorporating the distance metric. I'm particularly considering the discretised distance loss in this case which enables the KL divergence to work in a straight forward manner. The related work section at the end provides a nice overview of the literature and is very fair to other works in the space. I think contextualising the work in this manner should count towards originality, even if it does make the source of the ideas for the work more transparent.

## Quality
The experimental design and extensive nature of the experimentation is a large strength of this work for me. Particularly as the work is proposing a "framework" it is good to see such breadth of experimentation considered. The motivation and hypothesis of the work is clear and grounded well in the literature (to my knowledge). The results which are obtained do directly test the claims of the work and are interpreted fairly overall.

## Clarity
Overall the paper is well written, with clear tables and figures. The paper is structured well to support understanding and mathematical notation is clear, consistent and mostly intuitive. I also appreciate how the sections are structured in order of task complexity (as best they can be).

## Significance
Overall I think the work has the potential to inspire future work and does provide good results on the benchmarks. Once again the extensive literature review also supports the fact that this work has broad utility across a couple domains which supports its significance.

**Weaknesses:**

## Clarity
I find the manner that subsequences is introduces in Section 2.2 a bit unintuitive and requires some effort to parse. I'm puzzles that the subsequence needs to be sequential within the input sequence. I assume this is answered by the point: "...multiple elements are present within $s$, we limit our explanation to a singe $x$-subsequence here for clarity". I find that this is a bit too subtle of a statement to actual convey the fact that it could easily generalise (if indeed it can and so ease of explanation becomes the priority). For example, what would need to change to the formulation or equations to make it work for multiple sequences? What tasks are of this nature and why might a single sequence be a sufficient explanation? Does this require the permutation invariance of the LLM to work?

## Significance and Quality
I will ground these two sections as they share a common point. One of the primary issues for me is that the need for supervision on the distance metric is somewhat glossed over. Fundamentally the model is being given more supervision and so higher performance is expected. This limits significance to a degree, but this affect quality more for me as this should really be discussed. How easy it is to define metric spaces for a variety of problems is the determinant of the success and significance of the work and this should be more clearly acknowledged and discuss. The level of experimentation shown here does support that it is possible, but it is left to the reader to gauge and really this is where the conceptual insight of the work lies.

**Questions:**

I have listed a number of questions under weaknesses regarding the limitations of presenting the mathematical details using a single subsequence. I would appreciate if these could be answered.

---

> ### Author Response · Authors · 2025-11-17
>
> We thank the reviewer for acknowledging the originality of our framework and the importance of the experimental findings. Our responses are below.
>
> ---
>
> ## **1. Clarification on subsequence formulation**
>
> We agree that Section 2.2 can be explained more intuitively. Our intended formulation is as follows.
>
> **1. We define the element $x$ as the structured unit on which the metric is defined.**
> Examples include a bounding box $(x, y, w, h)$ or a set of robotic joint angles $(\theta_1, \theta_2, \theta_3)$, each of which belongs to a metric space $\mathcal{X}$ where distances such as Euclidean or angular distance are naturally defined.
>
> **2. For autoregressive modeling, we represent $x$ as a sequence of tokens $x = [x_i, \ldots, x_j]$.**
> Each component of the structured object becomes a token (for example, each coordinate, joint-angle component, or VQ index), yielding a multi-token subsequence that corresponds to the same underlying structured element $x$.
>
> **3. The loss for an element $x$ consists of the losses for the tokens in it.**
> In practice, we apply DIST$^2$Loss at each position $t \in {i, \ldots, j}$ by comparing the candidate value $v_t$ with the corresponding ground-truth value $x_t$.
> For example, if $x$ is a bounding box $(x, y, w, h)$ tokenized into four coordinate tokens, each coordinate position contributes its own distance-aware term; rotation parameters tokenized into multiple angle components or VQ codes tokenized into latent indices follow the same structure.
> This position-wise formulation is an efficient factorization of the object-level metric and is standard in autoregressive modeling.
> A fully joint metric over the entire multi-token element $x$ is conceptually straightforward but computationally infeasible: evaluating all multi-token alternatives would require comparing the full structured element with every possible configuration, scaling exponentially with the length of $x$. Because efficiency is a core motivation of DIST$^2$Loss, per-position decomposition preserves the intended geometric structure while remaining tractable.
> We only depart from this per-position formulation in the multi-token numeric case described in Appendix A.2.
>
> **4. When multiple structured elements appear in the input sequence $s$, we compute their losses independently.**
> Let the input contain structured elements $x^{(1)}, x^{(2)}, \ldots, x^{(K)}$, and let $\mathcal{T}^{(k)}$ denote the set of token positions corresponding to element $x^{(k)}$.
> The distance-based regularization term is:
>
> $L_{\text{dist}} = \sum_{k=1}^{K} \sum_{t \in T^{(k)}} \text{KL}( p_d^{(k)}(\cdot \mid x^{(k)}, t) \,\|\, p_{\theta}(\cdot \mid s_{<t}) )$
>
> Combined with the standard autoregressive cross-entropy:
>
> $L_{CE} = -\sum_{t=1}^{n} \log p_{\theta}(x_t | s_{<t})$
>
>
> we obtain the full objective:
>
> $L = L_{\text{CE}} + \alpha\ L_{\text{dist}}$
>
>
> Each structured element contributes an additive KL term, and the overall objective remains compatible with standard autoregressive training.
>
> We will revise Section 2.2 to reflect this more clearly.
>
> ---
>
> ### **2. On supervision and applicability**
>
> We agree that the applicability of this work depends on how often and how easily metric information can be obtained. DIST$^2$Loss only uses information that is already inherent in the task definition and requires no additional annotation, and we will clarify this more clearly in the revised draft.
>
> 1. **DIST$^2$Loss only exploits information readily available in the task definition and incurs no additional annotation.**
>
> The additional signal comes from structural properties of the output space that are already specified by the task, such as Euclidean distances for coordinates or latent-space distances for VQ tokens.
> This does not introduce new labels, auxiliary targets, or teacher models beyond the ground-truth sequences used in standard fine-tuning.
>
> 2. **Many practical tasks naturally provide such metric relationships.**
>
> A wide range of practical outputs have innate geometric structure together with natural distance functions.
> - Coordinate-based outputs in grounding, 2D and 3D localization, and robotic joint control use Euclidean distance in position space or angular distance for orientations.
> - Vector-quantized continuous outputs in image, audio, and video modeling inherit L2 or cosine distance from the codebook’s latent embedding space.
> - Scalar outputs such as rewards or function values naturally use absolute or squared differences.
> - Domain-specific outputs, for example in biological or molecular sequence modeling, often define edit distance or energy-based distances between structured states.
> While our experiments cover only a subset of these domains, such metric structures are widespread across non-linguistic tasks.
>
> We will refine the manuscript to make this scope and applicability explicit.

---

### Official Review · Reviewer_aRs1 · 2025-11-02

**Soundness:** 3
**Presentation:** 3
**Contribution:** 3
**Rating:** 6
**Confidence:** 3

**Summary:**

This paper focuses on the field of LLM extended to non-linguistic domains (e.g., multimodal understanding, robotic manipulation, generative reward modeling) and addresses the key problem that traditional one-hot targets and cross-entropy loss ignore the metric relationships (e.g., coordinates, rotation angles, quantized embeddings) inherent in tokens. Motivated by the inefficiency of conventional fine-tuning (neglecting metric structure) and the instability of RL methods (sampling/rollout noise), the authors aim to enhance model performance in low-data regimes while maintaining compatibility with existing architectures . The core method, DIST2Loss, transforms continuous exponential family distributions derived from inherent distance metrics (e.g., Euclidean distance, RMSE) into discrete categorical targets, computes distance-aware loss via KL divergence, and fuses it with standard cross-entropy loss. Empirical results show that DIST2Loss consistently improves performance across diverse tasks with the most notable gains in data-scarce settings .

**Strengths:**

1. The manuscript addresses a critical limitation of discrete autoregressive models (neglect of metric token relationships) with a principled solution—DIST2Loss directly embeds metric structure into the target distribution without relying on extra data or architectural modifications. This fills a gap in extending LLMs to non-linguistic tasks where spatial/numerical relationships matter.

2. DIST2Loss is validated on five distinct tasks (meta linear regression, visual grounding, robotic manipulation, reward modeling, image generation), demonstrating its broad applicability. This cross-task consistency strengthens the credibility of the method. It also performs well in low-data regimes.

**Weaknesses:**

1. Insufficient details on hyperparameter $\tau$: The temperature hyperparameter $\tau$  controls the smoothness of the target distribution, but the manuscript only states "small values for digits and larger values for VQ-VAE vocabularies" without providing specific values or a systematic sensitivity analysis.

2. I suggest the paper incorporates additional technical approaches for comparable distance perception methods. For instance, it could detail whether other methods achieve distance modeling through modifications to the loss function, adjustments to the target distribution, or the introduction of external modules.

3. Does the default setting of $\alpha$ = 0.1 potentially lead to a scale imbalance problem? Is it possible to incorporate an adaptive scaling mechanism to prevent this?

4. The authors evaluate other capabilities solely after single-task fine-tuning (e.g., MMLU after fine-tuning for reward modeling). However, they do not validate performance in multi-task fine-tuning scenarios.

5. The authors only validated general language abilities (MMLU) and did not cover non-linguistic fundamental abilities (e.g., image understanding capabilities). Including these aspects would significantly enhance the paper's quality.

**Questions:**

Please see Weaknesses.

---

> ### Author Response · Authors · 2025-11-17
>
> We thank the reviewer for recognizing that DIST2Loss provides a principled way to embed metric structure into discrete targets and for noting its consistent benefits across diverse tasks. We address the concerns below.
>
> ---
>
> ## **1. Details on temperature $\tau$**
>
> For decimal tokens, we set $\tau = 2.0$, which matches the squared Euclidean distances of the 1D digit lattice. This choice yields a likelihood kernel $\exp(-d^2/\tau)$ equivalent to a unit-variance Gaussian.
>
> VQ codebooks differ because similarity scales are not fixed: identical score gaps can correspond to different semantic distances depending on training, so the Gaussian analogy does not apply. Instead, we select $\tau$ using an information-theoretic criterion to maintain comparable entropy across vocabularies. A uniform distribution over size $V$ has entropy $H=\ln V$, so we set $\tau \approx 1/\ln V$ to preserve similar sharpness. For a VQ codebook with $V=16384$, this yields $\tau \approx 0.1$.
>
> This explanation is currently missing from the draft, and we will include it in the revision.
>
> ---
>
> ## **2. Related work expansion**
>
> We thank the reviewer for the suggestion. To address this, we will expand the related work by contrasting DIST$^2$Loss with existing distance-modelling approaches. A brief overview is as follows:
>
> **Loss**: Ordinal classification methods incorporate distance by modifying the loss function. Ordinal Label Distribution Learning [1] explicitly models distances between ordinal labels by learning a label distribution that reflects inter-label structure. Ordinal log-loss [2] similarly weights the log-loss term according to distances between labels. Such methods introduce task-specific loss formulations not readily compatible with LLMs.
>
> **Module**: Some works use additional components, such as using distance or graph relationships between inputs as priors for transformer attention mechanisms [3]. In contrast, DIST$^$Loss acts in the output space and therefore remains independent of architectural choices.
>
> **Target**: Non-uniform label smoothing explores target space. Class-Similarity Based Label Smoothing [4] assigns probability to classes in proportion to semantic similarity, and Instance-Based Label Smoothing [5] uses a teacher model’s output structure to distribute probability mass. DIST$^2$Loss differs in two important ways. First, it operates over the vocabulary of multimodal LLMs, enabling applications beyond image classification. Second, DIST$^2$Loss is targets training efficiency, whereas non-uniform label smoothing is primarily used for calibration.
>
>
> ---
>
> ## **3. Hyperparameter sweep on $\alpha$**
>
> We appreciate the reviewer’s suggestion and have conducted a hyperparameter sweep over the loss coefficient $\alpha$ in the reward modeling experiment. Specifically, we vary $\alpha$ to evaluate the sensitivity of DIST$^2$Loss to this parameter. The results indicate that the method is relatively robust across a wide range of values.
>
> | $\alpha$ | Accuracy (\%) |
> |:------------:|:--------|
> | 1.0     | 77,3   |
> | 0.2      | 77.8   |
> | **0.1**          | 85.3   |
> | 0.02          | 80.7   |
> | 0.01          | 79.9   |
> | 0.005         | 76.7   |
> | 0.001         | 75.6   |
> | SFT          | 75.4   |
>
> ---
>
> ## **4. Multi-task finetuning is important, but it is out-of-scope of the current work**
>
> Our focus is domain-specific specialization rather than large-scale multi-task finetuning, as our use cases involve adapting LLMs to targeted settings such as visual grounding or embodied agents. The MMLU evaluation after reward-modeling finetuning is included only to monitor knowledge retention.
>
> We agree that applying DIST$^2$Loss to multi-task learning is a promising direction that could yield more general-purpose, metric-aware models, but we did not pursue it here due to the substantial computational cost. We will highlight this as an important avenue for future work.
>
> ---
>
> ## **5. Catastrophic forgetting test on visual understanding task**
>
> We appreciate the suggestion. We evaluated the model fine-tuned with DIST$^2$Loss on RefCOCO (visual grounding) using an out-of-domain visual understanding benchmark, RealWorldQA. The results show that DIST$^2$Loss introduces only marginal forgetting on visual tasks as well.
>
>
>
> | model | Accuracy on ReadlWorldQA (\%) |
> |:------------:|:--------|
> | base (Phi-3.5V) | 54.4 |
> | DIST$^2$Loss on RefCOCO | 54.3 |
>
> ---
>
> ### **References**
>
>
> [1] Wen et al. "Ordinal label distribution learning." 2023.
>
> [2] Castagnos et al. "A simple log-based loss function for ordinal text classification." 2022.
>
> [3] Le et al. "Guiding visual question answering with attention priors." 2023.
>
> [4] Chihuang and JaJa. "Class-similarity based label smoothing for confidence calibration." 2021.
>
> [5] Maher and Kull. "Instance-based label smoothing for better calibrated classification networks." 2021.

---

### Author Response · Authors · 2025-12-01
**Summary of Reviewer Consensus & Response to Remaining Concerns**

To facilitate assessment of the review process, we summarize the overall reviewer landscape and our responses to the remaining concerns.


---


## **Main contribution of the paper**


- Introduces **DIST2Loss**, a training framework that **leverages metric structure** (e.g., Euclidean or angular distance) to **construct distance-aware target distributions** for discrete autoregressive models, without requiring architectural changes.
- Transforms continuous distance relationships into **discrete categorical targets**, allowing models to capture semantic proximity often ignored by standard cross-entropy loss (one-hot targets).
- Demonstrates consistent performance gains across **five diverse domains** (visual grounding, robotic manipulation, reward modeling, image generation, and regression), particularly in **low-data regimes**.


---


## **Overall reviewer stance**


**Three of four reviewers (aRs1, qCXL, Pvxr) support acceptance with positive scores. Notably, Reviewer Pvxr raised their score from 4 to 6 after clarifications regarding the scope of numerical reasoning.**

These reviewers praised the work for addressing a critical gap in non-linguistic token modeling, the breadth of the experimental validation, and the principled nature of the framework. They agreed that the method is original and effective across multiple domains.


---


## **Primary remaining disagreement (Reviewer 7BSi)**


Reviewer 7BSi maintained a negative score, arguing that the method lacks novelty and is equivalent to **Knowledge Distillation (KD)**.
- **Clarification on Novelty and KD**: We firmly established that this assessment relies on a misunderstanding of the method. **DIST2Loss is not Knowledge Distillation.** It utilizes *no teacher model, no soft logits, and no external supervision*. Instead, it derives targets deterministically from the ground truth and the task's intrinsic metric (e.g., spatial coordinates).
- **Technical Distinction**: While both methods use KL divergence, KD distills a teacher's learned distribution, whereas DIST2Loss enforces a geometric prior defined by the task. The other three reviewers explicitly recognized this originality.


---


## **Resolution of other concerns**


The positive reviewers raised valid technical questions which we addressed to their satisfaction:
- **Numerical Reasoning Scope (Reviewer Pvxr)**: The reviewer initially questioned if the model improved general arithmetic. We clarified that **DIST2Loss targets metric adherence within specific token spaces (e.g., coordinate proximity) rather than general mathematical reasoning.** We provided additional results on RefCOCO showing improved geometric alignment, leading the reviewer to raise their score. **We will refine the manuscript's framing to avoid overclaiming general numerical reasoning capabilities.**
- **Hyperparameter Selection (Reviewer aRs1)**: We provided the missing mathematical justification for the temperature $\tau$, showing how it is derived via entropy matching to ensure consistent target sharpness across different vocabulary sizes (e.g., VQ vs. digits).
- **Notation (Reviewer qCXL)**: We clarified the subsequence formulation to explain how the loss factorizes over token positions while preserving object-level metric structure.


---

## **Context for remaining disagreement**


The primary conflict arises from Reviewer 7BSi’s classification of the method as "Knowledge Distillation," which is factually incorrect as no teacher exists in our pipeline. The remaining three reviewers have reached a consensus that the method is sound, novel, and empirically strong, particularly after we clarified the scope of "metric awareness" versus "numerical reasoning."


---


We are happy to provide additional details if they would assist the AC’s assessment.

---

### Meta-Review · Area_Chair_oX5C · 2026-01-05

**Summary:**

Initial reviews for the paper were somewhat mixed, with 2 reviewers marginally arguing for acceptance, 1 marginally arguing for rejection, and 1 strong rejection.

Reviewer aRs1 was generally positive and largely asks for details on hyperparameter choice and questions on alternative experimental evaluation tasks.

Reviewer qCXL is also supportive, noting multiple positive aspects of the work.  The reviewer has a concern with the clarity of some of the technical presentation as well as questions over whether the method requires supervision to define an appropriate distance metric, which the authors clarify is not needed but comes from the inherent structure of the problems of interest.

Reviewer 7BSi is the most negative of the initial reviews and claims that the motivation and proposed method is not novel and has limited technical contribution.

Reviewer Pvxr initially votes for a marginal rejection before noting increasing their score to a marginal acceptance post-rebuttal.  The review reads rather positively (perhaps more than would be suggested by a score of 4 in fact), pointing out several strengths of the work, with the main critique appearing to be questions regarding numerical reasoning tasks.

Overall, while the initial reviews were somewhere between borderline-to-reject, I believe the authors have responded well to the questions raised by the reviewers and am fairly confident that given an appropriate discussion period the reviews would be revised to a consensus for acceptance.

**Reviewer Concerns:**

I believe reviewer aRs1's concerns were largely addressed/answered with further experimental results on hyperparameter choice and by noting that some of the proposed tasks are beyond the scope of the current work.

Likewise, qCXL's questions regarding clarity appear largely explained.  I would note that questions regarding whether the method requires supervision appear across multiple reviews, and thus the authors should take care to clarify this in a revised version, but this seems to be a relatively simple modification of writing/explanation.

I am convinced by the authors' response to 7BSi that the method operates in a distinct regime from the prior work on knowledge distillation, which the reviewer claimed limits the novelty of the current work.  This was the primary critique of reviewer 7BSi, so I consider the concerns resolved.

Reviewer Pvxr notes that they have a remaining unresolved question about the applicability of the method to numerical reasoning tasks, but I agree with the authors' response that this is somewhat outside the scope of the current work in that numerical reasoning requires one to first identify numeric structure within a general unstructured language query, whereas the current method is designed for problems where the numeric/geometric structure is inherently given as part of the problem.

**Reviewer Scores:**

The two positive reviews do not appear to have any outstanding issues, so I believe they would at least remain the same or (perhaps more likely) increase.  The marginally negative review (Pvxr) already notes increasing their score.  The most negative review (7BSi) notes high confidence in their initial score, but I am convinced by the authors' response, and I would require further justification from the reviewer if they wished to maintain their initial score.

---

### Decision · Program_Chairs · 2026-01-26

Accept (Poster)